# TabularBench: Benchmarking Adversarial Robustness for Tabular Deep Learning in Real-world Use-cases

**Thibault Simonetto**
University of Luxembourg
Luxembourg
thibault.simonetto@uni.lu

**Salah Ghamizi**
LIST / RIKEN AIP
Luxembourg
salah.ghamizi@gmail.com

**Maxime Cordy**
University of Luxembourg
Luxembourg
maxime.cordy@uni.lu

## Abstract

While adversarial robustness in computer vision is a mature research field, fewer researchers have tackled the evasion attacks against tabular deep learning, and even fewer investigated robustification mechanisms and reliable defenses. We hypothesize that this lag in the research on tabular adversarial attacks is in part due to the lack of standardized benchmarks. To fill this gap, we propose Tabular-Bench, the first comprehensive benchmark of robustness of tabular deep learning classification models. We evaluated adversarial robustness with CAA, an ensemble of gradient and search attacks which was recently demonstrated as the most effective attack against a tabular model. In addition to our open benchmark https://github.com/serval-uni-lu/tabularbench where we welcome submissions of new models and defenses, we implement 7 robustification mechanisms inspired by state-of-the-art defenses in computer vision and propose the largest benchmark of robust tabular deep-learning over 200 models across five critical scenarios in finance, healthcare, and security. We curated real datasets for each use case, augmented with hundreds of thousands of realistic synthetic inputs, and trained and assessed our models with and without data augmentations. We open-source our library that provides API access to all our pre-trained robust tabular models, and the largest datasets of real and synthetic tabular inputs. Finally, we analyze the impact of various defenses on the robustness and provide actionable insights to design new defenses and robustification mechanisms.

## 1 Introduction

Modern machine learning (ML) models have reached or surpassed human-level performance in numerous tasks, leading to their adoption in critical settings such as finance, security, and healthcare. However, concomitantly to their increasing deployment, researchers have uncovered significant vulnerabilities in generating valid adversarial examples (i.e., constraint-satisfying) where test or deployment data are manipulated to deceive the model. Most analyses of these performance drops have focused on the fields of Computer Vision and Large Language Models where extensive benchmarks for adversarial robustness are available (e.g., Croce et al. (2020) and Wang et al. (2023)).

Despite the widespread use of tabular data and the maturity of Deep Learning (DL) models for this field, the impact of evasion attacks on tabular data has not been thoroughly investigated. Although there are existing benchmarks for *in-distribution* (ID) tabular classification (Borisov et al., 2021), and distribution shifts (Gardner et al., 2023), there is no available benchmark of adversarial robustness for deep tabular models, in particular in critical real-world settings. We summarize in Table 1 these related benchmarks.

The need for dedicated benchmarks for tabular model robustness is enhanced by the unique challenges that tabular machine learning raises compared to computer vision and NLP tasks.

38th Conference on Neural Information Processing Systems (NeurIPS 2024) Track on Datasets and Benchmarks.

Table 1: Existing related benchmarks and their differences with ours

| Benchmark | Domain | Metric | Realistic evaluation |
|---|---|---|---|
| Tabsurvey (Borisov et al., 2021) | Tabular | ID performance | No |
| Tableshift (Gardner et al., 2023) | Tabular | OOD performance | No |
| ARES (Dong et al., 2020) | CV | Adversarial performance | No |
| Robustbench (Croce et al., 2020) | CV | Adversarial performance | Yes |
| DecodingTrust (Wang et al., 2023) | LLM | Trust (incl adversarial) | Yes |
| **OURS** | Tabular | Adversarial performance | Yes |

One significant challenge is that tabular data exhibit *feature constraints*, which are complex relationships and interactions between features. Satisfying these feature constraints can be a non-convex or even nondifferentiable problem, making established evasion attack algorithms relying on gradient descent ineffective in generating valid adversarial examples (i.e., constraint-satisfying) (Ghamizi et al., 2020). Furthermore, attacks designed specifically for tabular data often disregard feature-type constraints (Ballet et al., 2019) or, at best, consider categorical features without accounting for feature relationships (Wang et al., 2020; Xu et al., 2023; Bao et al., 2023), and are evaluated on datasets that contain only such features. This limitation restricts their applicability to domains with heterogeneous feature types.

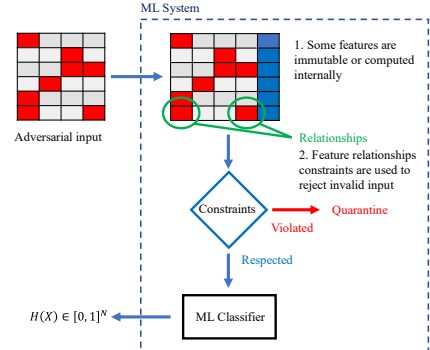

Figure 1: The main challenges for adversarial attacks in Tabular Machine Learning: When an adversary perturbs some features (red), it may not be aware of the new features that are computed internally and added (blue), or the relationships between features (green). If the monitoring system detects a constraint violation, the input is quarantined and a rejection (1) is returned.

Moreover, tabular ML models often involve specific feature engineering, that is, "secret" and inaccessible to an attacker. For example, in credit scoring applications, the end user can alter a subset of model features, but the other features result from internal processing that adds domain knowledge before reaching the model (Ghamizi et al., 2020). This raises the need for new threat models that take into account these specificities. We summarize the unique specificities of tabular machine learning and the challenges they pose to an adversarial user in Figure 1.

Thus, the machine learning research community currently lacks not only (1) an empirical understanding of the impact of architecture and robustification mechanisms on tabular data model architectures, but also (2) a reliable and high-quality benchmark to enable such investigations. Such a benchmark for tabular adversarial attacks should feature deployable attacks and defenses that reflect as accurately as possible the robustness of models within a reasonable computational budget. A reliable benchmark should also consider recent advances in tabular deep learning architectures and data augmentation techniques, and tackle realistic attack scenarios and real-world use cases considering their domain constraints and realistic capabilities of an attacker.

To address both gaps, we propose TabularBench, the first comprehensive benchmark of robustness of tabular deep learning classification models. We evaluated adversarial robustness using *Constrained Adaptive Attack (CAA)* (Simonetto et al., 2024), a combination of gradient-based and search-based attacks that have recently been shown to be the most effective against tabular models. We take advantage of our new benchmark and uncover unique findings on deep tabular learning architectures and defenses. We focus our study on defenses based on adversarial training (AT), and draw the following insights:

**Test performance is misleading:** Given the same tasks, different architectures have similar ID performance but lead to very disparate robust performances. Even more, data augmentations that improve ID performance can hurt robust performance.

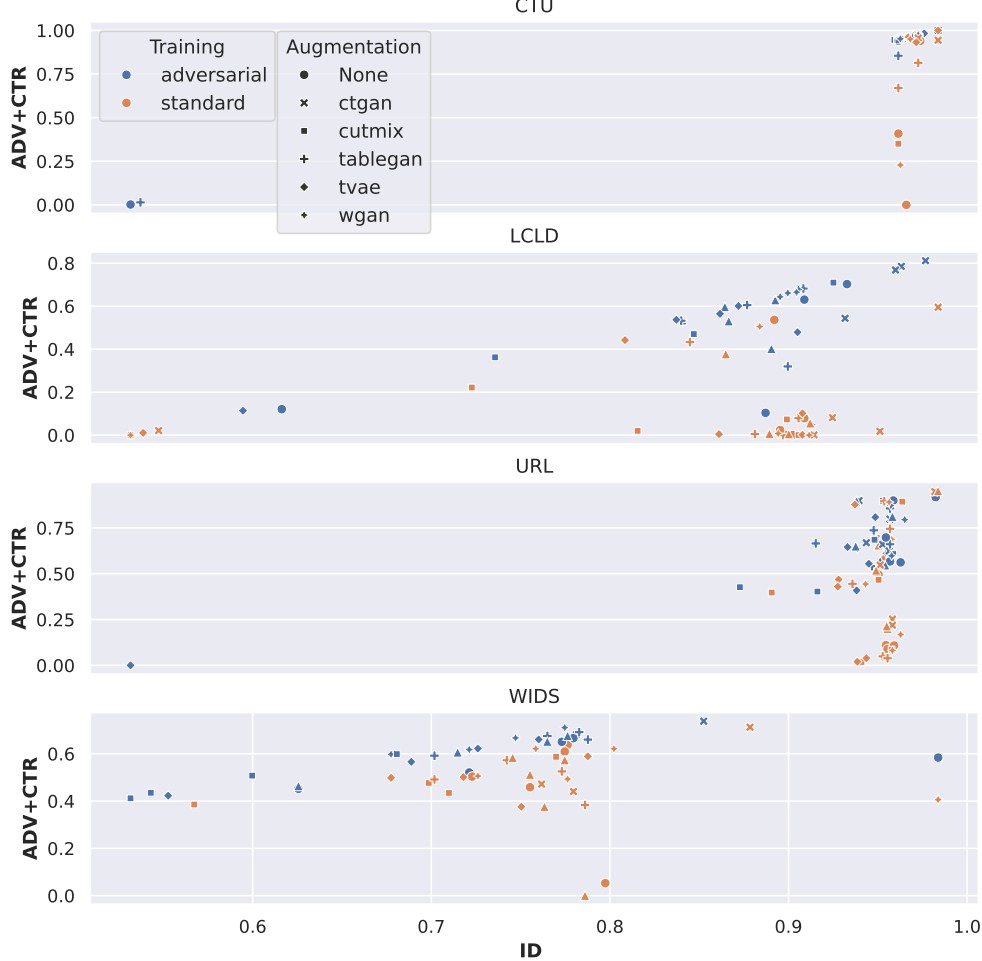

Figure 2: Summary of our main experiments; Y-axis: Robust Accuracy, X-axis ID accuracy

**Importance of domain constraints:** Disregarding domain constraints overestimates robustness and leads to selection of sub-optimal architectures and defenses when considering the domain constraints.

**Data augmentation effectiveness is task-specific.** There is no data augmentation that is optimal for both ID and robust performance across all tasks. Some simpler augmentations (like Cutmix) can outperform complex generative approaches.

**Contributions.** To summarize, our work makes the following key contributions:

- **Leaderboard** (`https://serval-uni-lu.github.io/tabularbench`): a website with a leaderboard based on *more than 200* evaluations to track the progress and the current state of the art in adversarial robustness of tabular deep learning models for each critical setting. The goal is to clearly identify the most successful ideas in tabular architectures and robust training mechanisms to accelerate progress in the field.

- **Dataset Zoo** : a collection of real and synthetic datasets generated with and without domain-constraint satisfaction, over five critical tabular machine learning use cases.

- **Model Zoo** : a collection of the most robust models that are easy to use for any downstream application. We pre-trained these models in particular on our five downstream tasks and we expect that this collection will promote the creation of more effective adversarial attacks by simplifying the evaluation process across a broad set of *over 200* models.

- **Analysis**: based on our trained models, we analyze how architectures, AT, and data augmentation mechanisms affect the robust performance of tabular deep learning models and provide insights on the best strategies per use case.

## 2  Background

Tabular data are one of the most common forms of data (Shwartz-Ziv and Armon, 2021), especially in critical applications such as medical diagnosis (Ulmer et al., 2020; Somani et al., 2021) and financial applications (Ghamizi et al., 2020; Cartella et al., 2021).

Traditional ML such as random forests and XGBoost often outperform DL on tabular data, primarily due to their robustness in handling feature heterogeneity and interdependence (Borisov et al., 2022).

To bridge the gap, researchers have proposed various improvements, from regularization mechanisms (e.g., RLN (Shavitt and Segal, 2018)) to attention layers (TabNet (Arik and Pfister, 2021)). These innovations are catching up and even outperforming shallow models in some settings, demonstrating the competitiveness of DL for Tabular Data.

The maturity of DL for ID tasks opens new perspectives for studying its performance in advanced settings, such as out-of-distribution (OOD) performance and adversarial robustness. One major work on OOD research is the Tableshift benchmark (Gardner et al., 2023), an exhaustive evaluation of the OOD performance of a variety of DNN classifiers. There is, however, to the best of our knowledge, no similar work on adversarial robustness, while the use cases when DL models are deployed for tabular data are among the most critical settings, and many are prone to malicious users.

Our work is the first exhaustive benchmark for the critical property of adversarial robustness of DL models. Our work is timely and leverages CAA (Simonetto et al., 2024), a novel attack previously demonstrated as the most effective and efficient tabular attack in the literature in multiple classification tasks under realistic constraints. CAA combines two attacks, CAPGD and MOEVA. CAPGD is an iterative gradient attack that maximizes the error and minimizes the features' constraint violations with regularization losses and projection mechanisms. MOEVA is a genetic algorithm attack that considers the three adversarial objectives: (1) classifier's error maximization, (2) perturbation minimization, and (3) constraint violations minimization, in its fitness function.

Although CAA was only evaluated against vanilla and simple madry AT, we have implemented advanced robustification mechanisms, inspired by proven techniques from top-performing research in the Robustbench computer vision benchmark Robustbench (Croce et al., 2020). Our work is the first implementation and evaluation of state-of-the-art defense mechanisms for tabular DL models.

## 3  TabularBench: Adversarial Robustness Benchmark for Tabular Data

In Appendix A.3 we report the detailed evaluation settings such as metrics, attack parameters, and hardware. We focus below on the datasets, classifiers, and synthetic data generators.

### 3.1  Tasks

We curated datasets meeting the following criteria: (1) **open source:** the datasets must be publicly available with a clear definition of the features and preprocessing, (2) **from real-world applications:** datasets that do not contain simulated data, (3) **binary classification:** datasets that support a meaningful binary classification task, and (4) **with feature relationships**: datasets that contain feature relationships and constraints, or they can be inferred directly from the definitions of features.

After an extensive review of tabular datasets, only the following five datasets match our requirements.

The **CTU** (Chernikova and Oprea, 2022) includes legitimate and botnet traffic from CTU University. Its challenge lies in the extensive number of linear domain constraints, totaling 360. **LCLD** (George, 2018) is a credit-scoring containing accepted and rejected credit requests. It has 28 features and 9 *non-linear* constraints. The most challenging dataset of our benchmark is the **Malware** dataset prepared by Dyrmishi et al. (2023). The very large number of features (24222), most of which are involved in each constraint, make this dataset challenging to attack. **URL** (Hannousse and Yahiouche, 2021) is a dataset comprising both legitimate and phishing URLs. Featuring only 14 linear domain constraints and 63 features, it represents the simplest case in our benchmark. The **WiDS** (Lee et al., 2020) includes medical data on the survival of patients admitted to the ICU, with only 31 linear domain constraints.

Our datasets include varying complexity in terms of number of features and constraints and diverse class imbalance intensity. We summarize the datasets and their relevant properties in Table 2 and provide more details in Appendix A.1 .

Table 2: Properties of the use cases of our benchmark.

| Dataset | Domain | Output to flip | Total size | # Features | # Ctrs | Inbalance |
|---------|--------|----------------|------------|------------|--------|-----------|
| CTU | Botnet detection | Malicious connections | 198 128 | 756 | 360 | 99.3/0.7 |
| LCLD | Credit scoring | Reject loan request | 1 220 092 | 28 | 9 | 80/20 |
| Malware | Malware detection | Malicious software | 17 584 | 24 222 | 7 | 45.5/54.5 |
| URL | Phishing | Malicious URL | 11 430 | 63 | 14 | 50/50 |
| WIDS | ICU survival | Expected survival | 91 713 | 186 | 31 | 91.4/8.6 |

## 3.2 Architectures

We consider five state-of-the-art deep tabular architectures from the survey by Borisov et al. (2021): **TabTransformer** (Huang et al., 2020) and **TabNet** (Arik and Pfister, 2021), are based on transformer architectures. **RLN** (Shavitt and Segal, 2018) uses a regularization coefficient to minimize a counterfactual loss, **STG** (Yamada et al., 2020) improves feature selection using stochastic gates, while **VIME** (Yoon et al., 2020) depends on self-supervised learning. We provide in Appendix A.2 the details of the architectures and the training hyperparameters. These architectures are on par with XGBoost, the top shallow machine-learning model for our applications.

## 3.3 Data Augmentation

Our benchmark considers synthetic data augmentation using five state-of-the-art tabular data generators. These generators were pre-trained to learn the distribution of the training data. Then, we augmented each of our datasets 100-fold (for example, for URL dataset, we generated 1.143.000 synthetic examples). Appendix A.4 details the generator architectures and the training hyperparameters.

**WGAN** (Arjovsky et al., 2017) is a typical generator-discriminator GAN model using Wasserstein loss. We follow the implementation of Stoian et al. (2024) and apply a MinMax transformation for continuous features and one-hot encoding for categorical to adapt this architecture for tabular data.

**TableGAN** (Park et al., 2018) is an improvement over standard GAN generators for tabular data. It adds a classifier (trained to learn the labels and feature relationships) to the generator-discriminator setup to improve semantic accuracy. TableGAN uses MinMax transformation for features.

**CTGAN** (Xu et al., 2019a) uses a conditional generator and training-by-sampling strategy in a generator-discriminator GAN architecture to model tabular data.

**TVAE** (Xu et al., 2019a) is an adaptation of the Variational AutoEncoder architecture for tabular data. It uses the same data transformations as CTGAN and training with ELBO loss.

**GOGGLE** (Liu et al., 2023) is a graph-based model that learns relational and functional dependencies in data using graphs and a message passing DNN, generating variables based on their neighborhood.

**Cutmix** (Yun et al., 2019) In computer vision, patches are cut and pasted among training images where the labels are also mixed proportionally. We adapted the approach to tabular ML and for each pair of rows of the same class, we randomly mix half of the features to generate a new sample.

For training, each batch of real examples is augmented with a same-size random synthetic batch (without replacement). However, the evaluation only runs on real examples. In AT, we generate adversarials from half of the real examples randomly selected and half of the synthetic examples.

## 3.4 Attack

To build our robustness benchmark, we leverage the Constrained Adaptive Attack (CAA) Simonetto et al. (2024) as the attack algorithm. To the best of our knowledge, CAA is the most effective and efficient tabular attack in the literature in multiple classification tasks under realistic constraints that appear in real-world applications. These constraints can be of four types: (1) mutability (whether a feature can be modified), (2) range (the minimum and maximum values a feature can take), (3) types

(the type of the feature, e.g., categorical or numerical), and (4) relations (the dependencies between features, e.g., the sum of two features must be equal to a third feature).

CAA is a novel attack that combines two attacks, Constrained Adaptive Projected Gradient Descent (CAPGD) and Multi-Objective Evolutionary Adversarial (MOEVA) attack.

We denote by $x \in \mathbb{R}^d$ an input example and by $y \in \{1, \ldots, C\}$ its correct label. Let $h : \mathbb{R}^d \to \mathbb{R}^C$ be a classifier and $h_{c_k}(x)$ the classification score that $h$ outputs for input $x$ to be in class $c_k$.

CAPGD (Simonetto et al., 2024) is an iterative attack that generates adversarial examples by computing the following perturbed example at each iteration:

$$z^{(k+1)} = P_{\mathcal{S}}\big(x^{(k)} + \eta^{(k)}(\nabla \mathcal{L}'(x^{(k)}))\big) \tag{1}$$
$$x^{(k+1)} = R_{\Omega}\Big(P_{\mathcal{S}}\big(x^{(k)} + \alpha \cdot (z^{(k+1)} - x^{(k)}) + (1 - \alpha) \cdot (x^{(k)} - x^{(k-1)})\big)\Big)$$

where $P_{\mathcal{S}}$ is the projection operator onto the set of maximum perturbation $\delta$ denoted $\mathcal{S}$, $R_{\Omega}$ is a repair operator for a subset of constraints $\Omega$, $\eta^{(k)}$ is the step size, $\alpha$ is the momentum parameter, and $\mathcal{L}'$ abbreviates the objective function to be maximized defined as:

$$\mathcal{L}'(x) = \mathcal{L}(x, y, h, \Omega) = l(h(x), y) - \sum_{\omega_i \in \Omega} penalty(x, \omega_i). \tag{2}$$

where $l$ is the loss function of the model, and $penalty$ is the penalty function for each relation constraint $\omega_i \in \Omega$.

MOEVA (Simonetto et al., 2022) multi-objective evolutionary algorithm based on NSGA-III that generates adversarial examples by minimizing the following objectives:

$$minimise \ g_1(x) \equiv h(x) \tag{3}$$
$$minimise \ g_2(x) \equiv L_p(x - x^0) \tag{4}$$
$$minimise \ g_3(x) \equiv \sum_{\omega_i \in \Omega} penalty(x, \omega_i) \tag{5}$$

where $x^0$ is the original input, $L_p$ is the $L_p$ norm, in our case $L_2$.

CAA only applies MOEVA when CAPGD fails to find an adversarial example. The attack is successful if it finds an adversarial example that is misclassified by the classifier and satisfies all the constraints.

Although to the extent of our knowledge, CAA is the best attack, we acknowledge that better attacks may be developed in the future. We provide the code for CAA in our repository, and we encourage the community to develop new attacks and evaluate them on our benchmark. Additionally, CAA is extendable in its design. Inspired by Auto-Attack, CAA is the sequential application of multiple strong attacks and complementary attacks, from fastest to slowest to find the best adversarial example. The attacks are complementary in the sense that they generate adversarial examples from different examples in the input space. This design allows for the easy integration of new attacks into the CAA framework. We encourage the development of new effective attacks and the evaluation of their complementarity with CAA.

### 3.5 TabularBench API

To encourage the wide adoption of TabularBench as the go-to place for Tabular Machine Learning evaluation, we designed its API to be modular, extensible, and standardized. We split its architecture into three independent components. More details of each component are provided in Appendix C.

**A dataset Zoo** For each dataset in this study, we have collected, cleaned, and pre-processed the existing raw dataset. We implemented a novel *Constraint Parser* where the user can write the relations in a natural human-readable format to describe the relationships between features. The processed datasets are loaded with a *Dataset factory*, then the user gets their associated meta-data and pre-defined constraints. The datasets are automatically downloaded when not found.

```
ds = dataset_factory.get_dataset("lcld_v2_iid")
metadata = ds.get_metadata(only_x=True)
constraints = ds.get_constraints()
```

**A model Zoo** Our API supports five architectures, and for each, six data augmentation techniques (as well as no data augmentation) and two training schemes (standard training and adversarial training). Hence, 70 pre-trained models for each of our five datasets are accessible. Below, we fine-tune with CAA AT and CTGAN augmentation a pre-trained Tabtransformer with Cutmix augmentation:

```
scaler = TabScaler(num_scaler="min_max", one_hot_encode=True)
scaler.fit(x, metadata["type"])
model = TabTransformer("regression", metadata, scaler=scaler,
                                      pretrained="LCLD_TabTr_Cutmix")
train_dataloader = CTGANDataLoader(dataset=ds, split="train", scaler=
                                      scaler, attack="caa")
model.fit(train_dataloader)
```

**A standarized benchmark** To generate our leaderboard, we offer a one-line command that loads a pre-trained model from the zoo, and reports the clean and robust accuracy of the model following our benchmark's setting (taking into consideration constraint satisfaction and L2 minimization):

```
clean_acc, robust_acc = benchmark(dataset='LCLD', model="TabTr_Cutmix"
                                  , distance='L2', constraints=True)
```

## 4 Empirical Findings

In the main paper, we provide multiple figures to visualize the main insights. We only report scenarios where data augmentation and adversarial training do not lead to performance collapse. We report in Appendix B all the results and investigate the collapsed scenarios.

### 4.1 Without Data Augmentations

We report the ID and robust accuracies of our architectures prior to data increase in Table 3.

Table 3: Clean and robust performances across all architectures in the form XX/YY. XX is the accuracy with standard training, and YY is the accuracy with adversarial training.

| Dataset | Accuracy | TabTr. | RLN | VIME | STG | TabNet |
|---------|----------|--------|-----|------|-----|--------|
| CTU | ID | 95.3/95.3 | 97.8/97.3 | 95.1/95.1 | 95.3/95.1 | 96.0/0.2 |
| | Robust | 95.3/95.3 | 94.1/97.1 | 40.8/94.0 | 95.3/95.1 | 0.0/0.2 |
| LCLD | ID | 69.5/73.9 | 68.3/69.5 | 67.0/65.5 | 66.4/15.6 | 67.4/0.0 |
| | Robust | 7.9/70.3 | 0.0/63.0 | 2.4/10.4 | 53.6/12.1 | 0.4/0.0 |
| MALWARE | ID | 95.0/95.0 | 95.0/96.0 | 95.0/92.0 | 93.0/93.0 | 99.0/99.0 |
| | Robust | 94.0/95.0 | 94.0/96.0 | 95.0/92.0 | 93.0/93.0 | 97.0/99.0 |
| URL | ID | 93.6/93.9 | 94.4/95.2 | 92.5/93.4 | 93.3/94.3 | 93.4/99.5 |
| | Robust | 8.9/56.7 | 10.8/56.2 | 49.5/69.8 | 58.0/90.0 | 11.0/91.8 |
| WIDS | ID | 75.5/77.3 | 77.5/78.0 | 72.3/72.1 | 77.7/62.6 | 79.8/98.4 |
| | Robust | 45.9/65.1 | 60.9/66.6 | 50.3/52.1 | 50.3/45.2 | 5.3/58.4 |

**All models on malware dataset are robust without data augmentation.** AT improves adversarial accuracy for all the cases, but AT alone is not sufficient to completely robustify the models on URL and WIDS datasets. All malware classification models are completely robust with and without adversarial training; hence, we will restrict the study of improved defenses with augmentation in the following sections to the remaining datasets.

### 4.2 Impact of Data Augmentations

**With data augmentation alone, ID and robust performances are not aligned.** In Figure 2 we study the impact of data augmentation on ID and robust performance, both in standard and adversarial

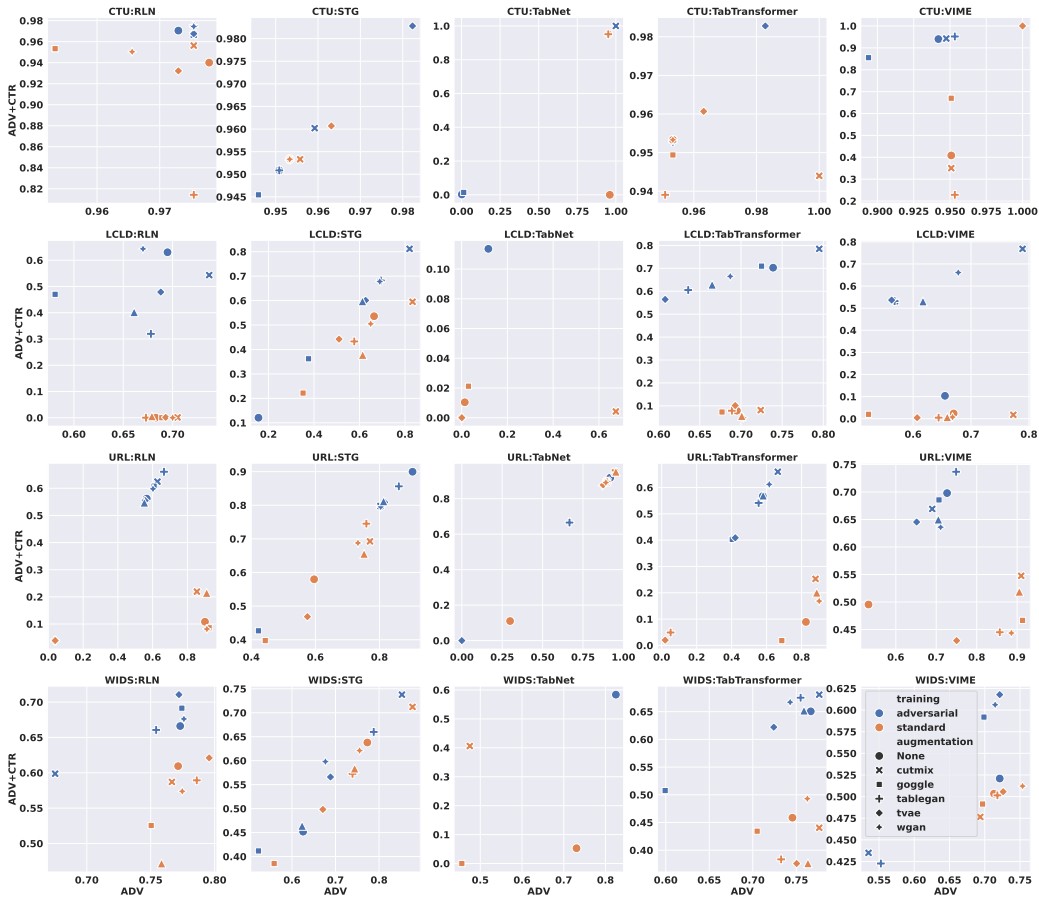

Figure 3: Robust performance while considering domain constraints (ADV+CTR: Y-axis) and without (ADV: X-axis) on all our use cases confirms the relevance of studying constrained-aware attacks.

training. With standard training, ID performance is misleading in CTU and URL datasets. Although all models exhibit similar ID performance, some of the augmentations lead to robust models, while others decrease it. CTGAN data augmentation is the best data augmentation for ID performance in all use cases, both with standard and adversarial training.

## 4.3 Impact of Adversarial Training

**With data augmentation and AT, ID and robust performances are correlated.** Although there is no trend of relationship between ID performance and robust performance in standard training, our study shows that robustness and ID performance are correlated after adversarial training. For example, the Pearson correlation between ID and robust performance increases from $0.15$ to $0.76$ for LCLD. All correlation values are in Appendix B.4.

Overall, all architectures can benefit from at least one data augmentation technique with adversarial training; however, standard training with data augmentation can outperform adversarial training without data augmentation (for e.g., on URL dataset using GOGGLE or CTGAN augmentations).

## 4.4 Impact of Architecture

In Figure 3 we study the robustness of each architecture with different defense mechanisms. We report both the robustness against unconstrained attacks (attacks unaware of domain knowledge) and attacks optimized to preserve the feature relationships and constraints.

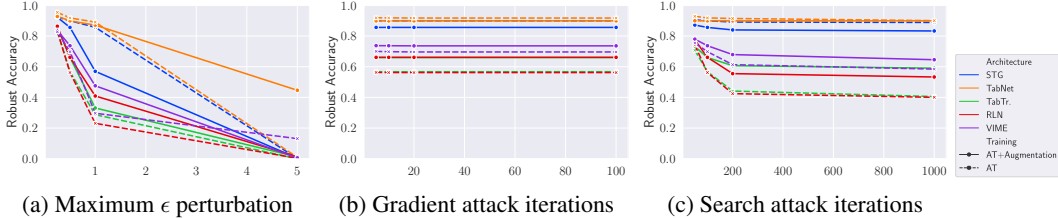

(a) Maximum $\epsilon$ perturbation  (b) Gradient attack iterations  (c) Search attack iterations

Figure 4: Impact of attack budget on the robust accuracy for URL dataset.

**Evaluation with unconstrained attacks is misleading.** Under standard training (orange scatters in Fig. 3), there is no relation between robustness to unconstrained attacks and the robustness when domain constraints are enforced. There is, however, a linear relationship under adversarial training with data augmentation only for STG, Tabstransformer, and VIME architectures. These results show that nonconstrained attacks are not sufficient to reliably assess the robustness of deep tabular models. Detailed correlation values are in the Appendix B.4.

**No data augmentation consistently outperforms the baselines with AT.** Among the 20 scenarios in Fig. 3, the original models achieve better constrained robustness than augmented models with adversarial training only for 4 scenarios: TabNet architecture on URL, LCLD and WIDS, and STG architecture on URL datasets. No data-augmentation technique consistently outperforms the others across all architectures. Cutmix, the simplest data augmentation, is often the best (in 7/20 scenarios).

### 4.5 Impact of Attack Budgets

We evaluated each robustified model against variants of the CAA attack, varying the $L_2$ distance of the perturbation $\epsilon$ from 0.5 to $\{0.25, 1, 5\}$, the gradient iterations from 10 to $\{5, 20, 100\}$, and the search iterations from 100 to $\{50, 200, 1000\}$. We report per architecture for each dataset the most robust model with AT and augmentation, and the robust model with AT only. We present in Fig. 4 the results for the URL dataset and refer to Appendix B.5 for the other use cases.

**AT+Augmentations models remain robust even under stronger attacks.** Our results show that the best defenses with AT+Augmentations (continuous lines) remain robust against increased gradient and search iteration budgets and remain more robust than AT alone (dashed lines) for VIME, RLN, and Tabtransformer architectures. Against an increase in perturbation size $\epsilon$, AT+Augmentations is more robust than AT alone for TabNet, TabTransformer, VIME, and RLN architectures. In particular, for $\epsilon = 5$, the robust accuracy of TabNet architectures remains above 40% with AT+Augmentations while the robust accuracy with AT alone drops to 0%.

## 5 Limitations

While our benchmark is the first to tackle adversarial robustness in tabular deep learning models, it does not cover all the directions of the field and focuses on domain constraints and defense mechanisms. Some of the orthogonal work is not addressed:

**Generalization to other distances:** We restricted our study to the $L_2$ distance to measure imperceptibility. Imperceptibility varies by domain, and several methods have been proposed to measure it (Ballet et al., 2019; Kireev et al., 2022; Dyrmishi et al., 2023). These methods have not been evaluated against human judgment or compared with one another, so there is no clear motivation to use one or another. In our research, we chose to use the well-established $L_2$ norm (following Dyrmishi et al. (2023)). Our algorithms and benchmarks support other distances and definitions of imperceptibility. We provide in Appendix B.6 an introduction to how our benchmark generalizes to other distances.

**Generalization to non-binary classification:** We restricted our study to binary tabular classification as it is the only case where we identified public datasets with domain constraints. The attacks used in our benchmark natively support multi-class classification. Our live leaderboard welcomes new datasets and will be updated if relevant datasets are designed by the community.

**Generalization to other types of defenses:** We only considered defenses based on data augmentation with adversarial training. Adversarial training-based defenses are recognized as the only reliable defenses against evasion attack (Tramer et al., 2020; Carlini, 2023). All other defenses are proven ineffective when the attacker is aware of them and performs adaptive attacks.

**Generalization to other (adaptive) attacks:** We only considered the Constrained Adaptive Attack (CAA) as it is the most effective and efficient attack in the literature. We encourage the community to develop new attacks and evaluate them on our benchmark. We provide the code for CAA in our repository and encourage the community to develop new attacks and evaluate them on our benchmark. In the face of new attacks, we will update our benchmark to include them. A limitation of our current evaluation regarding attacks is the lack of adaptive attacks, that adapt their strategy based on the defense mechanism. We welcome the development of new adaptive attacks and their evaluation on our benchmark at `https://github.com/serval-uni-lu/tabularbench/issues/new/choose`.

# 6 Broader Impact

Our work proposes the first benchmark of robustness of constrained tabular deep learning against evasion attacks. We focus on designing new defense mechanisms, inspired by effective approaches in computer vision (by combining data augmentation and adversarial training). Hence, we expect that our research will significantly contribute to the enhancement of defenses and will lead to even more resilient models, which may balance the potential harms research on adversarial attacks can have.

# Conclusion

In this work, we introduce TabularBench, the first benchmark of adversarial robustness of tabular deep learning models against constrained evasion attacks. We leverage Constrained Adaptive Attack (CAA), the best constrained tabular attack, to benchmark state-of-the-art architectures and defenses.

We provide a Python API to access the datasets, along with implementations of multiple tabular deep learning architectures, and provide all our pre-trained robust models directly through the API.

We conducted an empirical study that constitutes the first large-scale study of tabular data model robustness against evasion attacks. Our study covers five real-world use cases, five architectures, and six data augmentation mechanisms totaling more than 200 models. Our study identifies the best augmentation mechanisms for IID performance (CTGAN) and robust performance (Cutmix), and provides actionable insights on the selection of architectures and robustification mechanisms.

We are confident that our benchmark will accelerate the research of adversarial defenses for tabular ML and welcome all contributions to improve and extend our benchmark with new realistic use cases (multiclass), models, and defenses.

# Acknowledgments and Disclosure of Funding

This research was funded in whole, or in part, by the Luxembourg National Research Fund (FNR), grant reference NCER22/IS/16570468/NCER-FT and grant BRIDGES/2022/IS/17437536. This research was supported by BGL BNP Paribas Luxembourg. In particular, we would like to thank Anne Goujon and Andrey Boytsov for their support with the financial use case. The experiments presented in this paper were carried out using the HPC facilities of the University of Luxembourg Varrette et al. (2022) (see `hpc.uni.lu`). We would like to thank Salijona Dyrmishi for her help in generating synthetic examples using the data augmentation methods.

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

Table 4: The datasets evaluated in the empirical study, with the class imbalance of each dataset (Balance in %).

| Dataset | Task | Properties Size | # Features | Balance |
|---|---|---|---|---|
| LCLD (George, 2018) | Credit Scoring | 1 220 092 | 28 | 80/20 |
| CTU-13 (Chernikova and Oprea, 2022) | Botnet detection | 198 128 | 756 | 99.3/0.7 |
| URL (Hannousse and Yahiouche, 2021) | Phishing URL detection | 11 430 | 63 | 50/50 |
| WIDS (Lee et al., 2020) | ICU patient survival | 91 713 | 186 | 91.4/8.6 |

# A    Experimental protocol

## A.1    Datasets

Our dataset design followed the same protocol as Simonetto et al.Simonetto et al. (2022). We present in Table 4 the attributes of our datasets and the test performance achieved by each of the architectures.

**Credit Scoring - LCLD**    (license: CC0: Public Domain) We develop a dataset derived from the publicly accessible Lending Club Loan Data footnotehttps://www.kaggle.com/wordsforthewise/lending-club. This dataset includes 151 features, with each entry representing a loan approved by the Lending Club. However, some of these approved loans are not repaid and are instead charged off. Our objective is to predict, at the time of the request, whether the borrower will repay the loan or if it will be charged off. This dataset has been analyzed by various practitioners on Kaggle. Nevertheless, the original dataset only contains raw data, and to the best of our knowledge, there is no commonly used feature-engineered version. Specifically, caution is needed when reusing feature-engineered versions, as many proposed versions exhibit data leakage in the training set, making the prediction trivial. Therefore, we propose our own feature engineering. The original dataset contains 151 features. We exclude examples where the feature "loan status" is neither "Fully paid" nor "Charged Off," as these are the only definitive statuses of a loan; other values indicate an uncertain outcome. For our binary classifier, a "Fully paid" loan is represented as 0, and a "Charged Off" loan is represented as 1. We begin by removing all features that are missing in more than 30% of the examples in the training set. Additionally, we remove all features that are not available at the time of the loan request to avoid bias. We impute features that are redundant (e.g., grade and sub-grade) or too detailed (e.g., address) to be useful for classification. Finally, we apply one-hot encoding to categorical features. We end up with 47 input features and one target feature. We split the dataset using random sampling stratified by the target class, resulting in a training set of 915K examples and a testing set of 305K examples. Both sets are unbalanced, with only 20% of loans being charged off (class 1). We trained a neural network to classify accepted and rejected loans, consisting of 3 fully connected hidden layers with 64, 32, and 16 neurons, respectively. For each feature in this dataset, we define boundary constraints based on the extreme values observed in the training set. We consider the 19 features under the control of the Lending Club as immutable. We identify 10 relationship constraints (3 linear and 7 non-linear).

**URL Phishing - ISCX-URL2016**    (license CC BY 4.0) Phishing attacks are commonly employed to perpetrate cyber fraud or identity theft. These attacks typically involve a URL that mimics a legitimate one (e.g., a user's preferred e-commerce site) but directs the user to a fraudulent website that solicits personal or banking information. Hannousse and Yahiouche (2021) extracted features from both legitimate and fraudulent URLs, as well as external service-based features, to develop a classifier capable of distinguishing between fraudulent and legitimate URLs. The features extracted from the URL include the number of special substrings such as "www", "&", ";", "$", "and", the length of the URL, the port, the presence of a brand in the domain, subdomain, or path, and the inclusion of "http" or "https". External service-based features include the Google index, page rank, and the domain's presence in DNS records. The full list of features is available in the reproduction package. Hannousse and Yahiouche (2021) provide a dataset containing 5715 legitimate and 5715 malicious URLs. We use 75% of the dataset for training and validation, and the remaining 25% for testing and adversarial generation. We extract a set of 14 relational constraints between the URL features. Among these, 7 are linear constraints (e.g., the length of the hostname is less than or equal

Table 5: The three model architectures of our study.

| Family | Model | Hyperparameters |
|---|---|---|
| Transformer | TabTransformer | $hidden\_dim, n\_layers,$ $learning\_rate, norm, \theta$ |
| Transformer | TabNet | $n\_d, n\_steps,$ $\gamma, cat\_emb\_dim, n\_independent,$ $n\_shared, momentum, mask\_type$ |
| Regularization | RLN | $hidden\_dim, depth,$ $heads, weight\_decay,$ $learning\_rate, dropout$ |
| Regularization | STG | $hidden\_dims, learning\_rate, lam$ |
| Encoding | VIME | $p_m, \alpha, K, \beta$ |

to the length of the URL) and 7 are Boolean constraints of the form $if\ a > 0\ then\ b > 0$ (e.g., if the number of "http" > 0, then the number of slashes "/" > 0).

**Botnet attacks - CTU-13** (license CC BY NC SA 4.0) This is a feature-engineered version of CTU-13 proposed by Chernikova and Oprea (2019). It includes a combination of legitimate and botnet traffic flows from the CTU University campus. Chernikova et al. aggregated raw network data related to packets, duration, and bytes for each port from a list of commonly used ports. The dataset consists of 143K training examples and 55K testing examples, with 0.74% of examples labeled as botnet traffic (traffic generated by a botnet). The data contains 756 features, including 432 mutable features. We identified two types of constraints that define what constitutes feasible traffic data. The first type pertains to the number of connections and ensures that an attacker cannot reduce it. The second type involves inherent constraints in network communications (e.g., the maximum packet size for TCP/UDP ports is 1500 bytes). In total, we identified 360 constraints.

**WiDS** (license: PhysioNet Restricted Health Data License 1.5.0 [1]) Lee et al. (2020) dataset contains medical data on the survival of patients admitted to the ICU. The objective is to predict whether a patient will survive or die based on biological features (e.g., for triage). This highly unbalanced dataset has 30 linear relational constraints.

**Malware** (license MIT) contains 24222 features extracted from a collection of benign and malware Portable Executable (PE) files Dyrmishi et al. (2023). The features include the DLL imports, the API imports, PE sections, and statistic features such as the proportion of each possible byte value. The dataset contains 17,584 samples. The number of total features and the number of features involved in each constraint make this dataset challenging to attack. The objective of the classifier is to distinguish between malware and benign software.

## A.2 Model architectures

Table 5 provides an overview of the family, model architecture, and hyperparameters adjusted during the training of our models.

**TabTransformer** is a transformer-based model Huang et al. (2020). It employs self-attention to convert categorical features into an interpretable contextual embedding, which the paper asserts enhances the model's robustness to noisy inputs.

**TabNet** is another transformer-based model Arik and Pfister (2021). It utilizes multiple sub-networks in sequence. At each decision step, it applies sequential attention to select which features to consider. TabNet combines the outputs of each step to make the final decision.

**RLN** or Regularization Learning Networks Shavitt and Segal (2018) employs an efficient hyperparameter tuning method to minimize counterfactual loss. The authors train a regularization coefficient for the neural network weights to reduce sensitivity and create very sparse networks.

---

[1]https://physionet.org/content/widsdatathon2020/view-license/1.0.0/

**STG** or Stochastic Gates Yamada et al. (2020) uses stochastic gates for feature selection in neural network estimation tasks. The technique is based on a probabilistic relaxation of the $l_0$ norm of features or the count of selected features.

**VIME** or Value Imputation for Mask Estimation Yoon et al. (2020) employs self-supervised and semi-supervised learning through deep encoders and predictors.

### A.3 Evaluation settings

**Metrics** The models are fine-tuned to maximize cross-validation AUC. This metric is threshold-independent and is not affected by the class unbalance of our dataset.

We only attack clean examples that are not already misclassified by the model and from the critical class, that is respectively for each aforementioned dataset the class of phishing URLs, rejected loans, malwares, botnets, and not surviving patients. Because we consider a single class, the only relevant metric is robust accuracy on constrained examples. Unsuccessful adversarial examples count as correctly classified when measuring robust accuracy.

We only consider examples that respect domain constraints to compute robust accuracy. If an attack generates invalid examples, they are defacto considered unsuccessful and are reverted to their original example (correctly classified).

We report in the Appendix 8 all the remaining performance metrics, including the recall, the precision, and the Mattheu Correlation Coefficient (MCC).

**Attacks parameters** CAA applies CAPGD and MOEVA with the following parameters.

CAPGD uses $N_{iter} = 10$ iterations. The step reduction schedule for CPGD uses $M = 7$. In CAPGD, checkpoints are set as $w_j = \lceil p_j \times N_{iter} \rceil \leq N_{iter}$, with $p_j \in [0, 1]$ defined as $p_0 = 0$, $p_1 = 0.22$, and

$$p_{j+1} = p_j + \max p_j - p_{j-1} - 0.03, 0.06.$$

The influence of the previous update on the current update is set to $\alpha = 0.75$, and $\rho = 0.75$ for step halving. MOEVA runs for $n_{gen} = 100$ iterations, generating $n_{off} = 100$ offspring per iteration. Among the offspring, $n_{pop} = 200$ survive and are used for mating in the subsequent iteration.

**Hardware** Our experiments are conducted on an HPC cluster node equipped with 32 cores and 64GB of RAM allocated for our use. Each node is composed of 2 AMD Epyc ROME 7H12 processors running at 2.6 GHz, providing a total of 128 cores and 256 GB of RAM.

### A.4 Generator architectures

In our experimental study, we use the same five generative models as Stoian et al. (2024):

- **WGAN** (Arjovsky et al., 2017) is a GAN model trained with Wasserstein loss within a standard generator-discriminator GAN framework. In our implementation, WGAN utilizes a MinMax transformer for continuous features and one-hot encoding for categorical features. It is not specifically designed for tabular data.

- **TableGAN** (Park et al., 2018) is one of the pioneering GAN-based methods for generating tabular data. Besides the conventional generator and discriminator setup in GANs, the authors introduced a classifier trained to understand the relationship between labels and other features. This classifier ensures a higher number of semantically correct generated records. TableGAN applies a MinMax transformer to the features.

- **CTGAN** (Xu et al., 2019b) employs a conditional generator and a training-by-sampling strategy within a generator-discriminator GAN framework to model tabular data. The conditional generator produces synthetic rows conditioned on one of the discrete columns. The training-by-sampling method ensures that data are sampled according to the log frequency of each category, aiding in better modeling of imbalanced categorical columns. CTGAN uses one-hot encoding for discrete features and a mode-based normalization for continuous features. A variational Gaussian mixture model (**?**) is used to estimate the number of

modes and fit a Gaussian mixture. For each continuous value, a mode is sampled based on probability densities, and its mean and standard deviation are used for normalization.

- **TVAE** (Xu et al., 2019b) was introduced as a variant of the standard Variational AutoEncoder to handle tabular data. It employs the same data transformations as CTGAN and trains the encoder-decoder architecture using evidence lower-bound (ELBO) loss.

- **GOGGLE** (Liu et al., 2023) is a graph-based method for learning the relational structure of data as well as functional relationships (dependencies between features). The relational structure is learned by constructing a graph where nodes represent variables and edges indicate dependencies between them. Functional dependencies are learned through a message-passing neural network (MPNN). The generative model generates each variable considering its surrounding neighborhood.

The hyperpameters for training these models are based on Stoian et al. (2024) as well:

**For GOGGLE,** we employed the same optimizer and learning rate configuration as described in Liu et al. (2023). Specifically, ADAM was used with five different learning rates: $\{1 \times 10^{-3}, 5 \times 10^{-3}, 1 \times 10^{-2}\}$.

**For TVAE,** ADAM was utilized with five different learning rates: $\{5 \times 10^{-6}, 1 \times 10^{-5}, 1 \times 10^{-4}, 2 \times 10^{-4}, 1 \times 10^{-3}\}$.

For the other DGM models, three different optimizers were tested: ADAM, RMSPROP, and SGD, each with distinct sets of learning rates.

**For WGAN,** the learning rates were $\{1 \times 10^{-4}, 1 \times 10^{-3}\}$, $\{5 \times 10^{-5}, 1 \times 10^{-4}, 1 \times 10^{-3}\}$, and $\{1 \times 10^{-4}, 1 \times 10^{-3}\}$, respectively.

**For TableGAN,** the learning rates were $\{5 \times 10^{-5}, 1 \times 10^{-4}, 2 \times 10^{-4}, 1 \times 10^{-3}\}$, $\{1 \times 10^{-4}, 2 \times 10^{-4}, 1 \times 10^{-3}\}$, and $\{1 \times 10^{-4}, 1 \times 10^{-3}\}$, respectively.

**For CTGAN,** the learning rates were $\{5 \times 10^{-5}, 1 \times 10^{-4}, 2 \times 10^{-4}\}$, $\{1 \times 10^{-4}, 2 \times 10^{-4}, 1 \times 10^{-3}\}$, and $\{1 \times 10^{-4}, 1 \times 10^{-3}\}$, respectively.

For each optimizer-learning rate combination, three different batch sizes were tested, depending on the DGM model: $\{64, 128, 256\}$ for WGAN, $\{128, 256, 512\}$ for TableGAN, $\{70, 280, 500\}$ for CTGAN and TVAE, and $\{64, 128\}$ for GOGGLE. The batch sizes for CTGAN are multiples of 10 to accommodate the recommended PAC value of 10 as suggested in Lin et al. (2018), among other values.

### A.5 Reproduction package and availability

The source code, datasets, and pre-trained models required to replicate the experiments in this paper are publicly accessible under the MIT license on the repository `https://github.com/serval-uni-lu/tabularbench`.

## B Detailed results

### B.1 Baseline models performances

We compare in 6 the ID performance of XGBoost and our deep learning models under standard training. We confirm that DL models are on par with the performances achieved by shallow models.

### B.2 Execution time

We provide in Table 7 the execution time in seconds for each dataset and architecture. We run the benchmark on 1.000 examples each with the standard benchmark parameters (100 iterations, no time limit). Given the search component MOEVA within CAA, the execution time linearly increases with the complexity of the dataset. The malware dataset that we curated for this benchmark is very robust

Table 6: AUC In-distribution performance of models

| Dataset | CTU | LCLD | MALWARE | URL | WIDS |
|---------|-----|------|---------|-----|------|
| RLN | 0.991 | 0.719 | 0.993 | 0.984 | 0.869 |
| STG | 0.988 | 0.709 | 0.991 | 0.973 | 0.866 |
| TabNet | 0.996 | 0.722 | 0.994 | 0.986 | 0.870 |
| TabTr | 0.979 | 0.717 | 0.994 | 0.981 | 0.874 |
| VIME | 0.987 | 0.714 | 0.989 | 0.974 | 0.865 |
| XGBoost | 0.994 | 0.723 | 0.997 | 0.993 | 0.887 |

Table 7: Execution time of CAA in seconds, averaged over 5 seeds with 95 confidence interval.

| | TabTransformer | RLN | VIME | STG | TabNet |
|---|---|---|---|---|---|
| CTU | $110 \pm 5$ | $112 \pm 4$ | $116 \pm 2$ | $119 \pm 4$ | $182 \pm 4$ |
| LCLD | $83 \pm 2$ | $10 \pm 3$ | $13 \pm 1$ | $57 \pm 2$ | $23 \pm 0$ |
| MALWARE | $1509 \pm 125$ | $936 \pm 116$ | $3006 \pm 47$ | $971 \pm 103$ | $4008 \pm 220$ |
| URL | $17 \pm 1$ | $19 \pm 1$ | $51 \pm 2$ | $73 \pm 3$ | $58 \pm 1$ |
| WIDS | $49 \pm 1$ | $49 \pm 2$ | $41 \pm 1$ | $59 \pm 1$ | $25 \pm 1$ |

to CAA (see Table 3 of our benchmark) and extremely costly to attack. These properties make it a suitable use case to evaluate future attacks with our benchmark.

## B.3 Data augmentation detailed results

**Clean performance after data augmentation** We report in Table 8 the clean performances of our models under all the training scenarios. Notably, few training combinations lead to a collapse of performance ($MCC = 0$). It is the case on CTU dataset for all data augmentations with adversarial training, and CTGAN, Cutmix, and TVAE with standard training.

Table 8: Detailed results of clean performance for our augmented models

| Dataset | Arch | AUC | Accuracy | Precision | Recall | Mcc | Training | Augment |
|---------|------|-----|----------|-----------|--------|-----|----------|---------|
| URL | TabTr | 0.981 | 0.940 | 0.943 | 0.937 | 0.880 | Standard | None |
| URL | TabTr | 0.974 | 0.931 | 0.923 | 0.941 | 0.862 | Adversarial | None |
| URL | TabTr | 0.976 | 0.933 | 0.927 | 0.941 | 0.866 | Standard | ctgan |
| URL | TabTr | 0.963 | 0.916 | 0.903 | 0.932 | 0.832 | Adversarial | ctgan |
| URL | TabTr | 0.968 | 0.930 | 0.954 | 0.905 | 0.862 | Standard | cutmix |
| URL | TabTr | 0.956 | 0.900 | 0.937 | 0.857 | 0.803 | Adversarial | cutmix |
| URL | TabTr | 0.974 | 0.931 | 0.932 | 0.930 | 0.862 | Standard | goggle |
| URL | TabTr | 0.964 | 0.915 | 0.913 | 0.918 | 0.830 | Adversarial | goggle |
| URL | TabTr | 0.980 | 0.934 | 0.934 | 0.934 | 0.869 | Standard | wgan |
| URL | TabTr | 0.970 | 0.921 | 0.916 | 0.927 | 0.843 | Adversarial | wgan |
| URL | TabTr | 0.975 | 0.928 | 0.955 | 0.899 | 0.858 | Standard | tablegan |
| URL | TabTr | 0.967 | 0.919 | 0.935 | 0.900 | 0.839 | Adversarial | tablegan |
| URL | TabTr | 0.978 | 0.937 | 0.925 | 0.950 | 0.873 | Standard | tvae |
| URL | TabTr | 0.969 | 0.925 | 0.917 | 0.934 | 0.850 | Adversarial | tvae |
| URL | STG | 0.973 | 0.920 | 0.908 | 0.934 | 0.839 | Standard | None |
| URL | STG | 0.949 | 0.862 | 0.812 | 0.943 | 0.734 | Adversarial | None |
| URL | STG | 0.967 | 0.910 | 0.898 | 0.925 | 0.820 | Standard | ctgan |
| URL | STG | 0.959 | 0.895 | 0.863 | 0.940 | 0.794 | Adversarial | ctgan |
| URL | STG | 0.960 | 0.867 | 0.924 | 0.800 | 0.741 | Standard | cutmix |
| URL | STG | 0.954 | 0.842 | 0.909 | 0.760 | 0.694 | Adversarial | cutmix |
| URL | STG | 0.962 | 0.903 | 0.876 | 0.940 | 0.809 | Standard | goggle |
| URL | STG | 0.954 | 0.882 | 0.842 | 0.941 | 0.770 | Adversarial | goggle |
| URL | STG | 0.970 | 0.913 | 0.903 | 0.926 | 0.826 | Standard | wgan |

| | | | | | | | |
|---|---|---|---|---|---|---|---|
| URL | STG | 0.963 | 0.896 | 0.862 | 0.943 | 0.796 | Adversarial | wgan |
| URL | STG | 0.968 | 0.908 | 0.933 | 0.878 | 0.817 | Standard | tablegan |
| URL | STG | 0.956 | 0.888 | 0.862 | 0.923 | 0.777 | Adversarial | tablegan |
| URL | STG | 0.969 | 0.913 | 0.892 | 0.940 | 0.827 | Standard | tvae |
| URL | STG | 0.961 | 0.889 | 0.843 | 0.956 | 0.786 | Adversarial | tvae |
| URL | TabNet | 0.986 | 0.946 | 0.954 | 0.937 | 0.892 | Standard | None |
| URL | TabNet | 0.947 | 0.700 | 0.626 | 0.994 | 0.495 | Adversarial | None |
| URL | TabNet | 0.951 | 0.699 | 0.625 | 0.994 | 0.493 | Standard | ctgan |
| URL | TabNet | 0.943 | 0.853 | 0.819 | 0.905 | 0.709 | Adversarial | ctgan |
| URL | TabNet | 0.947 | 0.860 | 0.802 | 0.958 | 0.735 | Standard | cutmix |
| URL | TabNet | 0.935 | 0.860 | 0.815 | 0.934 | 0.729 | Adversarial | cutmix |
| URL | TabNet | 0.934 | 0.851 | 0.803 | 0.932 | 0.712 | Standard | goggle |
| URL | TabNet | 0.939 | 0.868 | 0.880 | 0.852 | 0.736 | Adversarial | goggle |
| URL | TabNet | 0.946 | 0.612 | 0.564 | 0.997 | 0.352 | Standard | wgan |
| URL | TabNet | 0.956 | 0.853 | 0.821 | 0.901 | 0.709 | Adversarial | wgan |
| URL | TabNet | 0.938 | 0.858 | 0.830 | 0.899 | 0.718 | Standard | tablegan |
| URL | TabNet | 0.929 | 0.504 | 1.000 | 0.008 | 0.063 | Adversarial | tablegan |
| URL | TabNet | 0.949 | 0.861 | 0.813 | 0.939 | 0.731 | Standard | tvae |
| URL | TabNet | 0.942 | 0.864 | 0.817 | 0.940 | 0.737 | Adversarial | tvae |
| URL | RLN | 0.984 | 0.945 | 0.945 | 0.946 | 0.891 | Standard | None |
| URL | RLN | 0.977 | 0.933 | 0.917 | 0.953 | 0.867 | Adversarial | None |
| URL | RLN | 0.980 | 0.939 | 0.938 | 0.941 | 0.878 | Standard | ctgan |
| URL | RLN | 0.973 | 0.925 | 0.914 | 0.939 | 0.851 | Adversarial | ctgan |
| URL | RLN | 0.983 | 0.944 | 0.945 | 0.942 | 0.887 | Standard | cutmix |
| URL | RLN | 0.977 | 0.933 | 0.924 | 0.944 | 0.866 | Adversarial | cutmix |
| URL | RLN | 0.978 | 0.938 | 0.937 | 0.940 | 0.877 | Standard | goggle |
| URL | RLN | 0.969 | 0.927 | 0.916 | 0.939 | 0.853 | Adversarial | goggle |
| URL | RLN | 0.982 | 0.940 | 0.945 | 0.934 | 0.880 | Standard | wgan |
| URL | RLN | 0.976 | 0.927 | 0.923 | 0.933 | 0.855 | Adversarial | wgan |
| URL | RLN | 0.980 | 0.934 | 0.953 | 0.913 | 0.868 | Standard | tablegan |
| URL | RLN | 0.971 | 0.925 | 0.933 | 0.915 | 0.850 | Adversarial | tablegan |
| URL | RLN | 0.982 | 0.941 | 0.939 | 0.944 | 0.883 | Standard | tvae |
| URL | RLN | 0.976 | 0.927 | 0.916 | 0.941 | 0.855 | Adversarial | tvae |
| URL | VIME | 0.974 | 0.928 | 0.929 | 0.927 | 0.856 | Standard | None |
| URL | VIME | 0.973 | 0.925 | 0.917 | 0.934 | 0.850 | Adversarial | None |
| URL | VIME | 0.968 | 0.916 | 0.906 | 0.927 | 0.831 | Standard | ctgan |
| URL | VIME | 0.965 | 0.912 | 0.913 | 0.911 | 0.824 | Adversarial | ctgan |
| URL | VIME | 0.971 | 0.922 | 0.921 | 0.924 | 0.844 | Standard | cutmix |
| URL | VIME | 0.967 | 0.918 | 0.915 | 0.921 | 0.836 | Adversarial | cutmix |
| URL | VIME | 0.960 | 0.900 | 0.908 | 0.891 | 0.801 | Standard | goggle |
| URL | VIME | 0.955 | 0.904 | 0.892 | 0.920 | 0.809 | Adversarial | goggle |
| URL | VIME | 0.968 | 0.917 | 0.913 | 0.923 | 0.835 | Standard | wgan |
| URL | VIME | 0.966 | 0.910 | 0.919 | 0.899 | 0.820 | Adversarial | wgan |
| URL | VIME | 0.963 | 0.905 | 0.930 | 0.875 | 0.811 | Standard | tablegan |
| URL | VIME | 0.960 | 0.906 | 0.923 | 0.887 | 0.813 | Adversarial | tablegan |
| URL | VIME | 0.968 | 0.914 | 0.919 | 0.907 | 0.828 | Standard | tvae |
| URL | VIME | 0.964 | 0.908 | 0.915 | 0.899 | 0.816 | Adversarial | tvae |
| LCLD | TabTr | 0.717 | 0.633 | 0.314 | 0.699 | 0.254 | Standard | None |
| LCLD | TabTr | 0.711 | 0.590 | 0.293 | 0.738 | 0.233 | Adversarial | None |
| LCLD | TabTr | 0.711 | 0.614 | 0.304 | 0.715 | 0.244 | Standard | ctgan |
| LCLD | TabTr | 0.694 | 0.526 | 0.271 | 0.803 | 0.212 | Adversarial | ctgan |
| LCLD | TabTr | 0.712 | 0.638 | 0.314 | 0.677 | 0.247 | Standard | cutmix |
| LCLD | TabTr | 0.702 | 0.596 | 0.294 | 0.723 | 0.230 | Adversarial | cutmix |
| LCLD | TabTr | 0.712 | 0.638 | 0.315 | 0.681 | 0.249 | Standard | goggle |
| LCLD | TabTr | 0.699 | 0.645 | 0.312 | 0.636 | 0.231 | Adversarial | goggle |
| LCLD | TabTr | 0.711 | 0.634 | 0.313 | 0.684 | 0.247 | Standard | wgan |
| LCLD | TabTr | 0.688 | 0.615 | 0.296 | 0.664 | 0.214 | Adversarial | wgan |
| LCLD | TabTr | 0.710 | 0.636 | 0.313 | 0.678 | 0.245 | Standard | tablegan |
| LCLD | TabTr | 0.694 | 0.651 | 0.313 | 0.614 | 0.225 | Adversarial | tablegan |

| | | | | | | | |
|------|-------|-------|-------|-------|-------|-------|-------------|----------|
| LCLD | TabTr | 0.716 | 0.634 | 0.314 | 0.693 | 0.252 | Standard | tvae |
| LCLD | TabTr | 0.702 | 0.620 | 0.304 | 0.691 | 0.235 | Adversarial | tvae |
| LCLD | STG | 0.709 | 0.646 | 0.317 | 0.660 | 0.245 | Standard | None |
| LCLD | STG | 0.679 | 0.788 | 0.432 | 0.172 | 0.170 | Adversarial | None |
| LCLD | STG | 0.705 | 0.503 | 0.266 | 0.841 | 0.215 | Standard | ctgan |
| LCLD | STG | 0.700 | 0.505 | 0.266 | 0.833 | 0.212 | Adversarial | ctgan |
| LCLD | STG | 0.707 | 0.766 | 0.404 | 0.347 | 0.231 | Standard | cutmix |
| LCLD | STG | 0.703 | 0.758 | 0.393 | 0.371 | 0.232 | Adversarial | cutmix |
| LCLD | STG | 0.704 | 0.677 | 0.331 | 0.591 | 0.242 | Standard | goggle |
| LCLD | STG | 0.698 | 0.616 | 0.300 | 0.687 | 0.229 | Adversarial | goggle |
| LCLD | STG | 0.705 | 0.669 | 0.326 | 0.606 | 0.241 | Standard | wgan |
| LCLD | STG | 0.699 | 0.657 | 0.318 | 0.617 | 0.234 | Adversarial | wgan |
| LCLD | STG | 0.702 | 0.710 | 0.349 | 0.509 | 0.238 | Standard | tablegan |
| LCLD | STG | 0.699 | 0.657 | 0.318 | 0.621 | 0.235 | Adversarial | tablegan |
| LCLD | STG | 0.706 | 0.652 | 0.319 | 0.645 | 0.244 | Standard | tvae |
| LCLD | STG | 0.706 | 0.625 | 0.307 | 0.687 | 0.239 | Adversarial | tvae |
| LCLD | TabNet | 0.722 | 0.656 | 0.326 | 0.668 | 0.262 | Standard | None |
| LCLD | TabNet | 0.656 | 0.799 | 0.000 | 0.000 | 0.000 | Adversarial | None |
| LCLD | TabNet | 0.687 | 0.785 | 0.270 | 0.042 | 0.031 | Standard | ctgan |
| LCLD | TabNet | 0.695 | 0.799 | 0.000 | 0.000 | 0.000 | Adversarial | ctgan |
| LCLD | TabNet | 0.700 | 0.799 | 1.000 | 0.000 | 0.003 | Standard | cutmix |
| LCLD | TabNet | 0.638 | 0.799 | 0.000 | 0.000 | 0.000 | Adversarial | cutmix |
| LCLD | TabNet | 0.673 | 0.799 | 0.000 | 0.000 | 0.000 | Standard | goggle |
| LCLD | TabNet | 0.683 | 0.201 | 0.201 | 1.000 | 0.000 | Adversarial | goggle |
| LCLD | TabNet | 0.665 | 0.799 | 0.000 | 0.000 | 0.000 | Standard | wgan |
| LCLD | TabNet | 0.688 | 0.799 | 0.000 | 0.000 | 0.000 | Adversarial | wgan |
| LCLD | TabNet | 0.689 | 0.793 | 0.255 | 0.016 | 0.015 | Standard | tablegan |
| LCLD | TabNet | 0.652 | 0.732 | 0.225 | 0.137 | 0.023 | Adversarial | tablegan |
| LCLD | TabNet | 0.667 | 0.799 | 0.248 | 0.000 | 0.002 | Standard | tvae |
| LCLD | TabNet | 0.696 | 0.799 | 0.000 | 0.000 | 0.000 | Adversarial | tvae |
| LCLD | RLN | 0.719 | 0.641 | 0.318 | 0.685 | 0.255 | Standard | None |
| LCLD | RLN | 0.716 | 0.628 | 0.309 | 0.693 | 0.245 | Adversarial | None |
| LCLD | RLN | 0.709 | 0.620 | 0.306 | 0.703 | 0.242 | Standard | ctgan |
| LCLD | RLN | 0.704 | 0.582 | 0.290 | 0.749 | 0.232 | Adversarial | ctgan |
| LCLD | RLN | 0.715 | 0.633 | 0.313 | 0.693 | 0.250 | Standard | cutmix |
| LCLD | RLN | 0.706 | 0.683 | 0.334 | 0.580 | 0.243 | Adversarial | cutmix |
| LCLD | RLN | 0.717 | 0.648 | 0.321 | 0.672 | 0.255 | Standard | goggle |
| LCLD | RLN | 0.710 | 0.644 | 0.317 | 0.666 | 0.247 | Adversarial | goggle |
| LCLD | RLN | 0.712 | 0.644 | 0.317 | 0.668 | 0.248 | Standard | wgan |
| LCLD | RLN | 0.705 | 0.646 | 0.316 | 0.653 | 0.241 | Adversarial | wgan |
| LCLD | RLN | 0.712 | 0.642 | 0.316 | 0.672 | 0.249 | Standard | tablegan |
| LCLD | RLN | 0.704 | 0.629 | 0.308 | 0.679 | 0.239 | Adversarial | tablegan |
| LCLD | RLN | 0.717 | 0.633 | 0.314 | 0.697 | 0.253 | Standard | tvae |
| LCLD | RLN | 0.708 | 0.635 | 0.312 | 0.676 | 0.244 | Adversarial | tvae |
| LCLD | VIME | 0.714 | 0.645 | 0.318 | 0.671 | 0.251 | Standard | None |
| LCLD | VIME | 0.713 | 0.651 | 0.321 | 0.657 | 0.250 | Adversarial | None |
| LCLD | VIME | 0.706 | 0.571 | 0.287 | 0.766 | 0.231 | Standard | ctgan |
| LCLD | VIME | 0.701 | 0.535 | 0.275 | 0.803 | 0.220 | Adversarial | ctgan |
| LCLD | VIME | 0.710 | 0.710 | 0.353 | 0.528 | 0.249 | Standard | cutmix |
| LCLD | VIME | 0.701 | 0.682 | 0.332 | 0.575 | 0.239 | Adversarial | cutmix |
| LCLD | VIME | 0.714 | 0.666 | 0.328 | 0.633 | 0.253 | Standard | goggle |
| LCLD | VIME | 0.703 | 0.685 | 0.334 | 0.569 | 0.239 | Adversarial | goggle |
| LCLD | VIME | 0.708 | 0.648 | 0.318 | 0.658 | 0.247 | Standard | wgan |
| LCLD | VIME | 0.699 | 0.660 | 0.320 | 0.618 | 0.237 | Adversarial | wgan |
| LCLD | VIME | 0.708 | 0.676 | 0.332 | 0.606 | 0.249 | Standard | tablegan |
| LCLD | VIME | 0.696 | 0.677 | 0.327 | 0.574 | 0.232 | Adversarial | tablegan |
| LCLD | VIME | 0.714 | 0.654 | 0.322 | 0.657 | 0.252 | Standard | tvae |
| LCLD | VIME | 0.705 | 0.628 | 0.308 | 0.684 | 0.240 | Adversarial | tvae |
| CTU | TabTr | 0.979 | 1.000 | 0.982 | 0.953 | 0.967 | Standard | None |

| | | | | | | | |
|---|---|---|---|---|---|---|---|
| CTU | TabTr | 0.985 | 1.000 | 0.982 | 0.953 | 0.967 | Adversarial | None |
| CTU | TabTr | 0.630 | 0.044 | 0.008 | 1.000 | 0.017 | Standard | ctgan |
| CTU | TabTr | 0.627 | 0.045 | 0.008 | 1.000 | 0.017 | Adversarial | ctgan |
| CTU | TabTr | 0.977 | 1.000 | 0.982 | 0.953 | 0.967 | Standard | cutmix |
| CTU | TabTr | 0.980 | 1.000 | 0.982 | 0.953 | 0.967 | Adversarial | cutmix |
| CTU | TabTr | 0.982 | 1.000 | 0.982 | 0.953 | 0.967 | Standard | wgan |
| CTU | TabTr | 0.984 | 1.000 | 0.982 | 0.953 | 0.967 | Adversarial | wgan |
| CTU | TabTr | 0.978 | 1.000 | 0.987 | 0.951 | 0.969 | Standard | tablegan |
| CTU | TabTr | 0.979 | 1.000 | 0.987 | 0.953 | 0.970 | Adversarial | tablegan |
| CTU | TabTr | 0.977 | 0.943 | 0.111 | 0.963 | 0.317 | Standard | tvae |
| CTU | TabTr | 0.974 | 0.609 | 0.018 | 0.983 | 0.103 | Adversarial | tvae |
| CTU | STG | 0.988 | 1.000 | 0.982 | 0.953 | 0.967 | Standard | None |
| CTU | STG | 0.986 | 1.000 | 0.992 | 0.951 | 0.971 | Adversarial | None |
| CTU | STG | 0.990 | 0.999 | 0.890 | 0.956 | 0.922 | Standard | ctgan |
| CTU | STG | 0.986 | 0.930 | 0.092 | 0.961 | 0.286 | Adversarial | ctgan |
| CTU | STG | 0.986 | 1.000 | 0.982 | 0.953 | 0.967 | Standard | cutmix |
| CTU | STG | 0.985 | 1.000 | 1.000 | 0.946 | 0.972 | Adversarial | cutmix |
| CTU | STG | 0.986 | 1.000 | 0.982 | 0.953 | 0.967 | Standard | wgan |
| CTU | STG | 0.985 | 1.000 | 0.982 | 0.953 | 0.967 | Adversarial | wgan |
| CTU | STG | 0.986 | 1.000 | 0.982 | 0.953 | 0.967 | Standard | tablegan |
| CTU | STG | 0.984 | 1.000 | 1.000 | 0.951 | 0.975 | Adversarial | tablegan |
| CTU | STG | 0.984 | 0.890 | 0.061 | 0.963 | 0.227 | Standard | tvae |
| CTU | STG | 0.981 | 0.436 | 0.013 | 0.983 | 0.072 | Adversarial | tvae |
| CTU | TabNet | 0.996 | 0.999 | 0.958 | 0.961 | 0.959 | Standard | None |
| CTU | TabNet | 0.978 | 0.993 | 0.500 | 0.002 | 0.035 | Adversarial | None |
| CTU | TabNet | 0.986 | 0.993 | 0.000 | 0.000 | 0.000 | Standard | ctgan |
| CTU | TabNet | 0.977 | 0.016 | 0.007 | 1.000 | 0.008 | Adversarial | ctgan |
| CTU | TabNet | 0.982 | 0.993 | 0.000 | 0.000 | 0.000 | Standard | cutmix |
| CTU | TabNet | 0.982 | 0.993 | 0.000 | 0.000 | 0.000 | Adversarial | cutmix |
| CTU | TabNet | 0.983 | 1.000 | 0.985 | 0.951 | 0.967 | Standard | wgan |
| CTU | TabNet | 0.987 | 0.993 | 0.000 | 0.000 | 0.000 | Adversarial | wgan |
| CTU | TabNet | 0.980 | 1.000 | 0.982 | 0.953 | 0.967 | Standard | tablegan |
| CTU | TabNet | 0.993 | 0.993 | 1.000 | 0.015 | 0.121 | Adversarial | tablegan |
| CTU | TabNet | 0.987 | 0.993 | 0.000 | 0.000 | 0.000 | Standard | tvae |
| CTU | TabNet | 0.976 | 0.007 | 0.007 | 1.000 | 0.000 | Adversarial | tvae |
| CTU | RLN | 0.991 | 0.998 | 0.819 | 0.978 | 0.894 | Standard | None |
| CTU | RLN | 0.990 | 0.999 | 0.904 | 0.973 | 0.937 | Adversarial | None |
| CTU | RLN | 0.994 | 0.986 | 0.338 | 0.975 | 0.570 | Standard | ctgan |
| CTU | RLN | 0.992 | 0.985 | 0.327 | 0.975 | 0.561 | Adversarial | ctgan |
| CTU | RLN | 0.989 | 1.000 | 0.987 | 0.953 | 0.970 | Standard | cutmix |
| CTU | RLN | 0.987 | 1.000 | 1.000 | 0.953 | 0.976 | Adversarial | cutmix |
| CTU | RLN | 0.991 | 0.999 | 0.887 | 0.966 | 0.925 | Standard | wgan |
| CTU | RLN | 0.990 | 0.999 | 0.923 | 0.975 | 0.949 | Adversarial | wgan |
| CTU | RLN | 0.992 | 0.999 | 0.880 | 0.975 | 0.926 | Standard | tablegan |
| CTU | RLN | 0.990 | 0.999 | 0.896 | 0.975 | 0.934 | Adversarial | tablegan |
| CTU | RLN | 0.988 | 0.987 | 0.362 | 0.973 | 0.589 | Standard | tvae |
| CTU | RLN | 0.988 | 0.986 | 0.338 | 0.975 | 0.570 | Adversarial | tvae |
| CTU | VIME | 0.987 | 1.000 | 0.997 | 0.951 | 0.974 | Standard | None |
| CTU | VIME | 0.983 | 1.000 | 0.997 | 0.951 | 0.974 | Adversarial | None |
| CTU | VIME | 0.972 | 0.007 | 0.007 | 1.000 | 0.000 | Standard | ctgan |
| CTU | VIME | 0.741 | 0.007 | 0.007 | 1.000 | 0.000 | Adversarial | ctgan |
| CTU | VIME | 0.991 | 1.000 | 0.997 | 0.951 | 0.974 | Standard | cutmix |
| CTU | VIME | 0.976 | 1.000 | 0.997 | 0.951 | 0.974 | Adversarial | cutmix |
| CTU | VIME | 0.977 | 1.000 | 1.000 | 0.953 | 0.976 | Standard | wgan |
| CTU | VIME | 0.979 | 1.000 | 0.997 | 0.953 | 0.975 | Adversarial | wgan |
| CTU | VIME | 0.984 | 1.000 | 0.997 | 0.951 | 0.974 | Standard | tablegan |
| CTU | VIME | 0.979 | 1.000 | 0.997 | 0.951 | 0.974 | Adversarial | tablegan |
| CTU | VIME | 0.950 | 0.008 | 0.007 | 1.000 | 0.001 | Standard | tvae |
| CTU | VIME | 0.727 | 0.007 | 0.007 | 1.000 | 0.000 | Adversarial | tvae |

| | | | | | | | |
|---|---|---|---|---|---|---|---|
| WIDS | TabTr | 0.874 | 0.810 | 0.287 | 0.755 | 0.383 | Standard | None |
| WIDS | TabTr | 0.869 | 0.794 | 0.272 | 0.772 | 0.373 | Adversarial | None |
| WIDS | TabTr | 0.868 | 0.799 | 0.279 | 0.780 | 0.383 | Standard | ctgan |
| WIDS | TabTr | 0.859 | 0.769 | 0.249 | 0.782 | 0.349 | Adversarial | ctgan |
| WIDS | TabTr | 0.866 | 0.835 | 0.314 | 0.708 | 0.395 | Standard | cutmix |
| WIDS | TabTr | 0.851 | 0.867 | 0.358 | 0.601 | 0.395 | Adversarial | cutmix |
| WIDS | TabTr | 0.873 | 0.805 | 0.285 | 0.784 | 0.392 | Standard | goggle |
| WIDS | TabTr | 0.853 | 0.784 | 0.261 | 0.764 | 0.357 | Adversarial | goggle |
| WIDS | TabTr | 0.866 | 0.797 | 0.273 | 0.763 | 0.371 | Standard | wgan |
| WIDS | TabTr | 0.864 | 0.788 | 0.264 | 0.764 | 0.361 | Adversarial | wgan |
| WIDS | TabTr | 0.869 | 0.808 | 0.284 | 0.748 | 0.378 | Standard | tablegan |
| WIDS | TabTr | 0.858 | 0.806 | 0.277 | 0.724 | 0.363 | Adversarial | tablegan |
| WIDS | TabTr | 0.871 | 0.801 | 0.280 | 0.776 | 0.383 | Standard | tvae |
| WIDS | TabTr | 0.858 | 0.790 | 0.264 | 0.747 | 0.356 | Adversarial | tvae |
| WIDS | STG | 0.866 | 0.782 | 0.260 | 0.776 | 0.361 | Standard | None |
| WIDS | STG | 0.865 | 0.875 | 0.381 | 0.627 | 0.424 | Adversarial | None |
| WIDS | STG | 0.852 | 0.638 | 0.183 | 0.878 | 0.285 | Standard | ctgan |
| WIDS | STG | 0.841 | 0.668 | 0.193 | 0.851 | 0.293 | Adversarial | ctgan |
| WIDS | STG | 0.863 | 0.885 | 0.400 | 0.567 | 0.414 | Standard | cutmix |
| WIDS | STG | 0.851 | 0.880 | 0.380 | 0.530 | 0.385 | Adversarial | cutmix |
| WIDS | STG | 0.851 | 0.780 | 0.253 | 0.742 | 0.342 | Standard | goggle |
| WIDS | STG | 0.837 | 0.727 | 0.218 | 0.787 | 0.310 | Adversarial | goggle |
| WIDS | STG | 0.863 | 0.800 | 0.274 | 0.744 | 0.366 | Standard | wgan |
| WIDS | STG | 0.855 | 0.855 | 0.334 | 0.625 | 0.384 | Adversarial | wgan |
| WIDS | STG | 0.861 | 0.846 | 0.326 | 0.676 | 0.396 | Standard | tablegan |
| WIDS | STG | 0.853 | 0.829 | 0.302 | 0.688 | 0.376 | Adversarial | tablegan |
| WIDS | STG | 0.857 | 0.776 | 0.252 | 0.758 | 0.345 | Standard | tvae |
| WIDS | STG | 0.845 | 0.807 | 0.271 | 0.678 | 0.341 | Adversarial | tvae |
| WIDS | TabNet | 0.870 | 0.777 | 0.259 | 0.796 | 0.365 | Standard | None |
| WIDS | TabNet | 0.835 | 0.104 | 0.090 | 0.984 | 0.003 | Adversarial | None |
| WIDS | TabNet | 0.853 | 0.090 | 0.090 | 1.000 | 0.000 | Standard | ctgan |
| WIDS | TabNet | 0.863 | 0.090 | 0.090 | 1.000 | 0.000 | Adversarial | ctgan |
| WIDS | TabNet | 0.866 | 0.910 | 0.000 | 0.000 | 0.000 | Standard | cutmix |
| WIDS | TabNet | 0.859 | 0.090 | 0.090 | 1.000 | 0.000 | Adversarial | cutmix |
| WIDS | TabNet | 0.856 | 0.090 | 0.090 | 1.000 | 0.000 | Standard | goggle |
| WIDS | TabNet | 0.862 | 0.090 | 0.090 | 1.000 | 0.000 | Adversarial | goggle |
| WIDS | TabNet | 0.865 | 0.795 | 0.275 | 0.787 | 0.381 | Standard | wgan |
| WIDS | TabNet | 0.855 | 0.090 | 0.090 | 1.000 | 0.000 | Adversarial | wgan |
| WIDS | TabNet | 0.864 | 0.090 | 0.090 | 1.000 | 0.000 | Standard | tablegan |
| WIDS | TabNet | 0.860 | 0.090 | 0.090 | 1.000 | 0.000 | Adversarial | tablegan |
| WIDS | TabNet | 0.857 | 0.104 | 0.090 | 0.984 | 0.003 | Standard | tvae |
| WIDS | TabNet | 0.864 | 0.090 | 0.090 | 1.000 | 0.000 | Adversarial | tvae |
| WIDS | RLN | 0.869 | 0.796 | 0.274 | 0.774 | 0.376 | Standard | None |
| WIDS | RLN | 0.867 | 0.789 | 0.268 | 0.779 | 0.370 | Adversarial | None |
| WIDS | RLN | 0.862 | 0.788 | 0.264 | 0.761 | 0.360 | Standard | ctgan |
| WIDS | RLN | 0.425 | 0.090 | 0.090 | 1.000 | 0.000 | Adversarial | ctgan |
| WIDS | RLN | 0.870 | 0.802 | 0.280 | 0.769 | 0.381 | Standard | cutmix |
| WIDS | RLN | 0.859 | 0.834 | 0.307 | 0.681 | 0.379 | Adversarial | cutmix |
| WIDS | RLN | 0.864 | 0.797 | 0.276 | 0.774 | 0.378 | Standard | goggle |
| WIDS | RLN | 0.857 | 0.777 | 0.256 | 0.782 | 0.358 | Adversarial | goggle |
| WIDS | RLN | 0.866 | 0.782 | 0.260 | 0.774 | 0.359 | Standard | wgan |
| WIDS | RLN | 0.858 | 0.770 | 0.249 | 0.776 | 0.347 | Adversarial | wgan |
| WIDS | RLN | 0.868 | 0.773 | 0.254 | 0.785 | 0.356 | Standard | tablegan |
| WIDS | RLN | 0.860 | 0.797 | 0.273 | 0.760 | 0.370 | Adversarial | tablegan |
| WIDS | RLN | 0.868 | 0.776 | 0.259 | 0.803 | 0.367 | Standard | tvae |
| WIDS | RLN | 0.854 | 0.756 | 0.237 | 0.774 | 0.332 | Adversarial | tvae |
| WIDS | VIME | 0.865 | 0.823 | 0.298 | 0.721 | 0.384 | Standard | None |
| WIDS | VIME | 0.858 | 0.817 | 0.291 | 0.720 | 0.376 | Adversarial | None |
| WIDS | VIME | 0.482 | 0.090 | 0.090 | 1.000 | 0.000 | Standard | ctgan |

| WIDS | VIME | 0.482 | 0.090 | 0.090 | 1.000 | 0.000 | Adversarial | ctgan |
|------|------|-------|-------|-------|-------|-------|-------------|-------|
| WIDS | VIME | 0.857 | 0.833 | 0.309 | 0.697 | 0.387 | Standard | cutmix |
| WIDS | VIME | 0.849 | 0.878 | 0.374 | 0.543 | 0.385 | Adversarial | cutmix |
| WIDS | VIME | 0.849 | 0.812 | 0.280 | 0.700 | 0.358 | Standard | goggle |
| WIDS | VIME | 0.840 | 0.802 | 0.268 | 0.700 | 0.346 | Adversarial | goggle |
| WIDS | VIME | 0.861 | 0.796 | 0.270 | 0.753 | 0.365 | Standard | wgan |
| WIDS | VIME | 0.845 | 0.791 | 0.259 | 0.715 | 0.339 | Adversarial | wgan |
| WIDS | VIME | 0.864 | 0.828 | 0.305 | 0.716 | 0.389 | Standard | tablegan |
| WIDS | VIME | 0.853 | 0.882 | 0.388 | 0.553 | 0.399 | Adversarial | tablegan |
| WIDS | VIME | 0.858 | 0.808 | 0.280 | 0.726 | 0.367 | Standard | tvae |
| WIDS | VIME | 0.846 | 0.787 | 0.256 | 0.721 | 0.339 | Adversarial | tvae |

For LCLD dataset only Goggle and WGAN data augmentations lead to $MCC = 0$. To uncover what happens with some generated data, we study the distribution of artificial examples on the LCLD dataset for 3 cases: Two cases where performance did not collapse: TableGAN and CTGAN and one problematic case WGAN.

**Kernel Density Estimation.** We first compare the artificial examples distributions in Figure 5. The results show that the labels and the main features of TableGAN, a "healthy" generator are closer to the distribution of the "problematic" generator WGAN than to the distribution of CTGAN, another "healthy" generator. Feature and label distributions are not problematic.

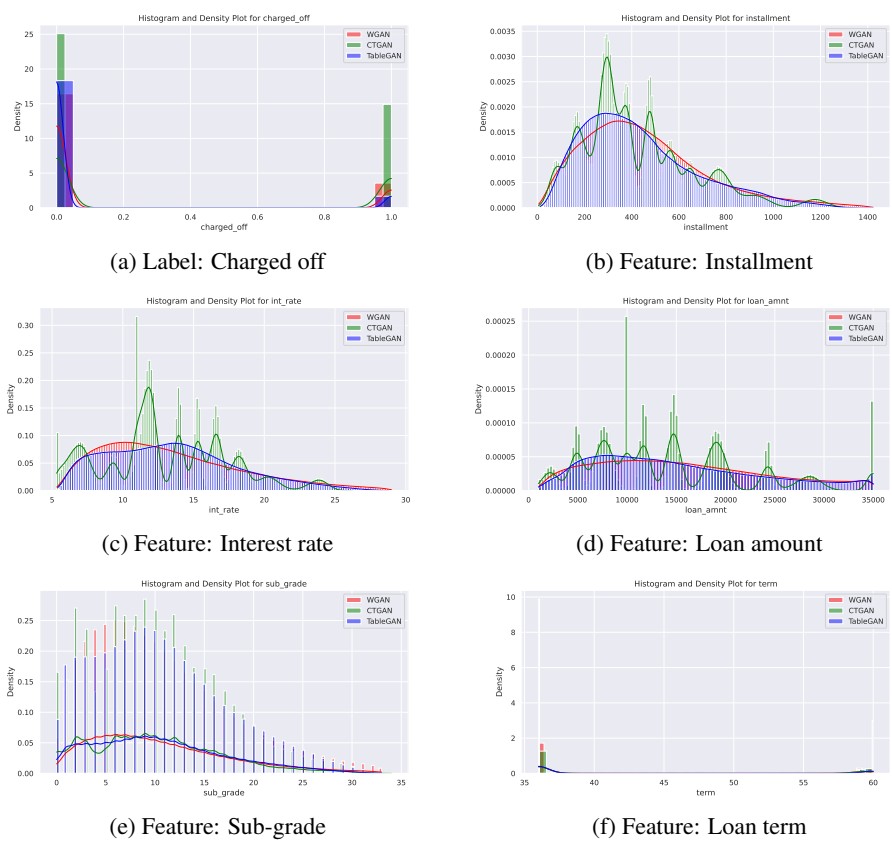

(a) Label: Charged off

(b) Feature: Installment

(c) Feature: Interest rate

(d) Feature: Loan amount

(e) Feature: Sub-grade

(f) Feature: Loan term

Figure 5: Impact of attack budget on the robust accuracy for LCLD dataset.

**Statistical analysis.** We perform the following statistical tests to compare the distributions quantitatively between the examples generated by the three generators. Kolmogorov-Smirnov test, t-test, or MWU test. We report the results in Table 9. Across all statistical tests, there is no specific pattern to the faulty generator "WGAN" compared to CTGAN and TableGAN.

Table 9: Statistical tests between the distributions of the 3 generators: W:WGAN, T:TableGAN, C:CTGAN, MWU:Mann-Whitney U.

| GAN | Test | Amount | Term | Rate | Installment | Sub-grade | Label |
|---|---|---|---|---|---|---|---|
| (W/T) | KS Statistic | 0.047 | 0.120 | 0.055 | 0.046 | 0.031 | 0.095 |
| (W/T) | KS p-value | 0.000 | 0.000 | 0.000 | 0.000 | 0.000 | 0.000 |
| (W/T) | t-test Statistic | 35.923 | 10.782 | -7.687 | 40.512 | 0.224 | 140.654 |
| (W/T) | t-test p-value | 0.000 | 0.000 | 0.000 | 0.000 | 0.823 | 0.000 |
| (W/T) | MWU Statistic | $1.3 \times 10^{11}$ | $1.2 \times 10^{11}$ | $1.2 \times 10^{11}$ | $1.3 \times 10^{11}$ | $1.2 \times 10^{11}$ | $1.3 \times 10^{11}$ |
| (W/T) | MWU p-value | 0.000 | 0.000 | 0.000 | 0.000 | 0.000 | 0.000 |
| (W/C) | KS Statistic | 0.112 | 0.056 | 0.105 | 0.089 | 0.037 | 0.194 |
| (W/C) | KS p-value | 0.000 | 0.000 | 0.000 | 0.000 | 0.000 | 0.000 |
| (W/C) | t-test Statistic | 80.112 | -21.286 | 40.896 | 61.097 | 30.043 | -221.351 |
| (W/C) | t-test p-value | 0.000 | 0.000 | 0.000 | 0.000 | 0.000 | 0.000 |
| (W/C) | MWU Statistic | $1.3 \times 10^{11}$ | $1.2 \times 10^{11}$ | $1.2 \times 10^{11}$ | $1.3 \times 10^{11}$ | $1.2 \times 10^{11}$ | $9.8 \times 10^{10}$ |
| (W/C) | MWU p-value | 0.000 | 0.002 | 0.000 | 0.000 | 0.000 | 0.000 |
| (T/C) | KS Statistic | 0.079 | 0.070 | 0.093 | 0.044 | 0.027 | 0.289 |
| (T/C) | KS p-value | 0.000 | 0.000 | 0.000 | 0.000 | 0.000 | 0.000 |
| (T/C) | t-test Statistic | -43.986 | 31.467 | -51.028 | -20.991 | -30.376 | 364.250 |
| (T/C) | t-test p-value | 0.000 | 0.000 | 0.000 | 0.000 | 0.000 | 0.000 |
| (T/C) | MWU Statistic | $1.2 \times 10^{11}$ | $1.2 \times 10^{11}$ | $1.2 \times 10^{11}$ | $1.2 \times 10^{11}$ | $1.2 \times 10^{11}$ | $1.6 \times 10^{11}$ |
| (T/C) | MWU p-value | 0.000 | 0.000 | 0.000 | 0.000 | 0.000 | 0.000 |

**Classification performance.** We build a new classifier to identify examples generated by WGAN and by TableGAN. We leverage Oodeel[2], a library that performs post-hoc deep OOD (Out-of-Distribution) detection.

The classifier reaches achieves a random accuracy (0.5) confirming that no specific features are sufficient to distinguish both generators.

Next, we evaluate the Maximum Logit Score (MLS) detector and report the histograms and AUROC curve of the detector in Figure 6.

Both the ROC curves and the histograms confirm that WGAN and TableGAN are not distinguishable.

**Conclusion:** From all our analysis, we confirm that the collapse of performance of training with WGAN data augmentation is not due to some evident properties in the generated examples.

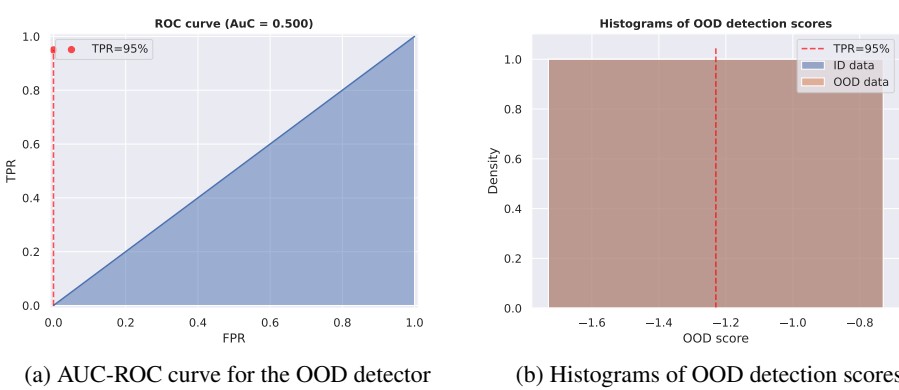

(a) AUC-ROC curve for the OOD detector      (b) Histograms of OOD detection scores

Figure 6: Performance of the OOD detector on the WGAN samples.

---

[2]https://github.com/deel-ai/oodeel

**Robust performance after data augmentation**   We report below the robustness of our 270 models trained with various combinations of arhcitecture, data augmentation, and adversarial training.

Table 10: Detailed results of adversarial robustness with constrained (CTR) and unconstrained attacks (ADV) across our 5 seeds. Adv. means adversarial training, Std. means standard training.

| Dataset | Arch | Training | Augment | Mean | | | Std | | |
|---------|------|----------|---------|------|-----|-----|-----|-----|-----|
| | | | | ID | CTR | ADV | ID | CTR | ADV |
| CTU | STG | Adv. | None | 0.951 | 0.951 | 0.951 | 0.000 | 0.000 | 0.000 |
| CTU | STG | Adv. | ctgan | 0.961 | 0.960 | 0.959 | 0.000 | 0.001 | 0.002 |
| CTU | STG | Adv. | cutmix | 0.946 | 0.945 | 0.946 | 0.000 | 0.001 | 0.000 |
| CTU | STG | Adv. | tablegan | 0.951 | 0.951 | 0.951 | 0.000 | 0.000 | 0.000 |
| CTU | STG | Adv. | tvae | 0.983 | 0.983 | 0.982 | 0.000 | 0.000 | 0.001 |
| CTU | STG | Adv. | wgan | 0.953 | 0.953 | 0.953 | 0.000 | 0.000 | 0.000 |
| CTU | STG | Std. | None | 0.953 | 0.953 | 0.953 | 0.000 | 0.000 | 0.000 |
| CTU | STG | Std. | ctgan | 0.956 | 0.953 | 0.956 | 0.000 | 0.000 | 0.000 |
| CTU | STG | Std. | cutmix | 0.953 | 0.953 | 0.953 | 0.000 | 0.000 | 0.000 |
| CTU | STG | Std. | tablegan | 0.953 | 0.953 | 0.953 | 0.000 | 0.000 | 0.000 |
| CTU | STG | Std. | tvae | 0.963 | 0.961 | 0.963 | 0.000 | 0.000 | 0.000 |
| CTU | STG | Std. | wgan | 0.953 | 0.953 | 0.953 | 0.000 | 0.000 | 0.000 |
| CTU | TabNet | Adv. | None | 0.002 | 0.002 | 0.002 | 0.000 | 0.001 | 0.001 |
| CTU | TabNet | Adv. | ctgan | 1.000 | 1.000 | 1.000 | 0.000 | 0.000 | 0.000 |
| CTU | TabNet | Adv. | cutmix | 0.000 | 0.000 | 0.000 | 0.000 | 0.000 | 0.000 |
| CTU | TabNet | Adv. | tablegan | 0.015 | 0.014 | 0.014 | 0.000 | 0.001 | 0.001 |
| CTU | TabNet | Adv. | tvae | 1.000 | 1.000 | 1.000 | 0.000 | 0.000 | 0.000 |
| CTU | TabNet | Adv. | wgan | 0.000 | 0.000 | 0.000 | 0.000 | 0.000 | 0.000 |
| CTU | TabNet | Std. | None | 0.961 | 0.000 | 0.961 | 0.000 | 0.000 | 0.000 |
| CTU | TabNet | Std. | ctgan | 0.000 | 0.000 | 0.000 | 0.000 | 0.000 | 0.000 |
| CTU | TabNet | Std. | cutmix | 0.000 | 0.000 | 0.000 | 0.000 | 0.000 | 0.000 |
| CTU | TabNet | Std. | tablegan | 0.953 | 0.953 | 0.953 | 0.000 | 0.000 | 0.000 |
| CTU | TabNet | Std. | tvae | 0.000 | 0.000 | 0.000 | 0.000 | 0.000 | 0.000 |
| CTU | TabNet | Std. | wgan | 0.951 | 0.951 | 0.951 | 0.000 | 0.000 | 0.000 |
| CTU | TabTr | Adv. | None | 0.953 | 0.953 | 0.953 | 0.000 | 0.000 | 0.000 |
| CTU | TabTr | Adv. | ctgan | 1.000 | 0.944 | 1.000 | 0.000 | 0.010 | 0.000 |
| CTU | TabTr | Adv. | cutmix | 0.953 | 0.953 | 0.953 | 0.000 | 0.000 | 0.000 |
| CTU | TabTr | Adv. | tablegan | 0.953 | 0.953 | 0.953 | 0.000 | 0.001 | 0.000 |
| CTU | TabTr | Adv. | tvae | 0.983 | 0.983 | 0.983 | 0.000 | 0.000 | 0.000 |
| CTU | TabTr | Adv. | wgan | 0.953 | 0.953 | 0.953 | 0.000 | 0.000 | 0.000 |
| CTU | TabTr | Std. | None | 0.953 | 0.953 | 0.953 | 0.000 | 0.000 | 0.000 |
| CTU | TabTr | Std. | ctgan | 1.000 | 0.944 | 1.000 | 0.000 | 0.005 | 0.000 |
| CTU | TabTr | Std. | cutmix | 0.953 | 0.949 | 0.953 | 0.000 | 0.003 | 0.000 |
| CTU | TabTr | Std. | tablegan | 0.951 | 0.939 | 0.951 | 0.000 | 0.001 | 0.000 |
| CTU | TabTr | Std. | tvae | 0.963 | 0.961 | 0.963 | 0.000 | 0.000 | 0.000 |
| CTU | TabTr | Std. | wgan | 0.953 | 0.953 | 0.953 | 0.000 | 0.000 | 0.000 |
| CTU | RLN | Adv. | None | 0.973 | 0.971 | 0.973 | 0.000 | 0.000 | 0.000 |
| CTU | RLN | Adv. | ctgan | 0.975 | 0.967 | 0.975 | 0.000 | 0.001 | 0.000 |
| CTU | RLN | Adv. | cutmix | 0.953 | 0.953 | 0.953 | 0.000 | 0.000 | 0.000 |
| CTU | RLN | Adv. | tablegan | 0.975 | 0.975 | 0.975 | 0.000 | 0.001 | 0.000 |
| CTU | RLN | Adv. | tvae | 0.975 | 0.968 | 0.975 | 0.000 | 0.002 | 0.000 |
| CTU | RLN | Adv. | wgan | 0.975 | 0.974 | 0.975 | 0.000 | 0.001 | 0.000 |
| CTU | RLN | Std. | None | 0.978 | 0.940 | 0.978 | 0.000 | 0.003 | 0.000 |
| CTU | RLN | Std. | ctgan | 0.975 | 0.956 | 0.975 | 0.000 | 0.002 | 0.000 |
| CTU | RLN | Std. | cutmix | 0.953 | 0.953 | 0.953 | 0.000 | 0.000 | 0.000 |
| CTU | RLN | Std. | tablegan | 0.975 | 0.814 | 0.975 | 0.000 | 0.026 | 0.000 |
| CTU | RLN | Std. | tvae | 0.973 | 0.932 | 0.973 | 0.000 | 0.011 | 0.000 |
| CTU | RLN | Std. | wgan | 0.966 | 0.950 | 0.966 | 0.000 | 0.001 | 0.000 |
| CTU | VIME | Adv. | None | 0.951 | 0.940 | 0.942 | 0.000 | 0.005 | 0.006 |

| CTU | VIME | Adv. | ctgan | 1.000 | 1.000 | 1.000 | 0.000 | 0.000 | 0.000 |
|------|------|------|----------|-------|-------|-------|-------|-------|-------|
| CTU | VIME | Adv. | cutmix | 0.951 | 0.943 | 0.947 | 0.000 | 0.004 | 0.002 |
| CTU | VIME | Adv. | tablegan | 0.951 | 0.855 | 0.894 | 0.000 | 0.016 | 0.008 |
| CTU | VIME | Adv. | tvae | 1.000 | 1.000 | 1.000 | 0.000 | 0.000 | 0.000 |
| CTU | VIME | Adv. | wgan | 0.953 | 0.952 | 0.953 | 0.000 | 0.001 | 0.000 |
| CTU | VIME | Std. | None | 0.951 | 0.408 | 0.951 | 0.000 | 0.049 | 0.000 |
| CTU | VIME | Std. | ctgan | 1.000 | 1.000 | 1.000 | 0.000 | 0.000 | 0.000 |
| CTU | VIME | Std. | cutmix | 0.951 | 0.350 | 0.951 | 0.000 | 0.029 | 0.000 |
| CTU | VIME | Std. | tablegan | 0.951 | 0.670 | 0.951 | 0.000 | 0.021 | 0.000 |
| CTU | VIME | Std. | tvae | 1.000 | 1.000 | 1.000 | 0.000 | 0.000 | 0.000 |
| CTU | VIME | Std. | wgan | 0.953 | 0.229 | 0.953 | 0.000 | 0.022 | 0.000 |
| LCLD | STG | Adv. | None | 0.156 | 0.121 | 0.156 | 0.000 | 0.001 | 0.000 |
| LCLD | STG | Adv. | ctgan | 0.820 | 0.812 | 0.820 | 0.000 | 0.001 | 0.000 |
| LCLD | STG | Adv. | cutmix | 0.376 | 0.362 | 0.376 | 0.000 | 0.000 | 0.000 |
| LCLD | STG | Adv. | goggle | 0.694 | 0.682 | 0.694 | 0.000 | 0.000 | 0.000 |
| LCLD | STG | Adv. | tablegan | 0.627 | 0.601 | 0.627 | 0.000 | 0.001 | 0.000 |
| LCLD | STG | Adv. | tvae | 0.689 | 0.678 | 0.689 | 0.000 | 0.000 | 0.000 |
| LCLD | STG | Adv. | wgan | 0.613 | 0.597 | 0.613 | 0.000 | 0.000 | 0.000 |
| LCLD | STG | Std. | None | 0.664 | 0.536 | 0.664 | 0.000 | 0.001 | 0.000 |
| LCLD | STG | Std. | ctgan | 0.833 | 0.595 | 0.833 | 0.000 | 0.004 | 0.000 |
| LCLD | STG | Std. | cutmix | 0.352 | 0.222 | 0.352 | 0.000 | 0.002 | 0.000 |
| LCLD | STG | Std. | goggle | 0.577 | 0.433 | 0.577 | 0.000 | 0.002 | 0.000 |
| LCLD | STG | Std. | tablegan | 0.510 | 0.442 | 0.510 | 0.000 | 0.001 | 0.000 |
| LCLD | STG | Std. | tvae | 0.649 | 0.505 | 0.649 | 0.000 | 0.001 | 0.000 |
| LCLD | STG | Std. | wgan | 0.614 | 0.377 | 0.614 | 0.000 | 0.002 | 0.000 |
| LCLD | TabNet | Adv. | None | 0.000 | 0.000 | 0.001 | 0.000 | 0.000 | 0.000 |
| LCLD | TabNet | Adv. | ctgan | 0.000 | 0.000 | 0.001 | 0.000 | 0.000 | 0.000 |
| LCLD | TabNet | Adv. | cutmix | 0.000 | 0.000 | 0.001 | 0.000 | 0.000 | 0.000 |
| LCLD | TabNet | Adv. | goggle | 1.000 | 1.000 | 1.000 | 0.000 | 0.000 | 0.000 |
| LCLD | TabNet | Adv. | tablegan | 0.116 | 0.114 | 0.117 | 0.000 | 0.000 | 0.000 |
| LCLD | TabNet | Adv. | tvae | 0.000 | 0.000 | 0.001 | 0.000 | 0.000 | 0.000 |
| LCLD | TabNet | Adv. | wgan | 0.000 | 0.000 | 0.001 | 0.000 | 0.000 | 0.000 |
| LCLD | TabNet | Std. | None | 0.674 | 0.004 | 0.674 | 0.000 | 0.001 | 0.000 |
| LCLD | TabNet | Std. | ctgan | 0.029 | 0.021 | 0.030 | 0.000 | 0.001 | 0.000 |
| LCLD | TabNet | Std. | cutmix | 0.000 | 0.000 | 0.001 | 0.000 | 0.000 | 0.000 |
| LCLD | TabNet | Std. | goggle | 0.000 | 0.000 | 0.001 | 0.000 | 0.000 | 0.000 |
| LCLD | TabNet | Std. | tablegan | 0.013 | 0.010 | 0.014 | 0.000 | 0.001 | 0.000 |
| LCLD | TabNet | Std. | tvae | 0.000 | 0.000 | 0.001 | 0.000 | 0.000 | 0.000 |
| LCLD | TabNet | Std. | wgan | 0.000 | 0.000 | 0.001 | 0.000 | 0.000 | 0.000 |
| LCLD | TabTr | Adv. | None | 0.739 | 0.703 | 0.739 | 0.000 | 0.001 | 0.000 |
| LCLD | TabTr | Adv. | ctgan | 0.795 | 0.785 | 0.795 | 0.000 | 0.001 | 0.000 |
| LCLD | TabTr | Adv. | cutmix | 0.725 | 0.710 | 0.725 | 0.000 | 0.001 | 0.000 |
| LCLD | TabTr | Adv. | goggle | 0.636 | 0.605 | 0.636 | 0.000 | 0.002 | 0.000 |
| LCLD | TabTr | Adv. | tablegan | 0.608 | 0.564 | 0.608 | 0.000 | 0.003 | 0.000 |
| LCLD | TabTr | Adv. | tvae | 0.687 | 0.665 | 0.687 | 0.000 | 0.001 | 0.000 |
| LCLD | TabTr | Adv. | wgan | 0.665 | 0.628 | 0.665 | 0.000 | 0.002 | 0.000 |
| LCLD | TabTr | Std. | None | 0.695 | 0.079 | 0.695 | 0.000 | 0.006 | 0.000 |
| LCLD | TabTr | Std. | ctgan | 0.724 | 0.081 | 0.724 | 0.000 | 0.004 | 0.000 |
| LCLD | TabTr | Std. | cutmix | 0.677 | 0.073 | 0.677 | 0.000 | 0.008 | 0.000 |
| LCLD | TabTr | Std. | goggle | 0.689 | 0.079 | 0.689 | 0.000 | 0.004 | 0.000 |
| LCLD | TabTr | Std. | tablegan | 0.693 | 0.101 | 0.693 | 0.000 | 0.005 | 0.000 |
| LCLD | TabTr | Std. | tvae | 0.703 | 0.048 | 0.703 | 0.000 | 0.003 | 0.000 |
| LCLD | TabTr | Std. | wgan | 0.701 | 0.055 | 0.701 | 0.000 | 0.005 | 0.000 |
| LCLD | RLN | Adv. | None | 0.695 | 0.630 | 0.695 | 0.000 | 0.001 | 0.000 |
| LCLD | RLN | Adv. | ctgan | 0.737 | 0.543 | 0.737 | 0.000 | 0.001 | 0.000 |
| LCLD | RLN | Adv. | cutmix | 0.581 | 0.470 | 0.581 | 0.000 | 0.003 | 0.000 |
| LCLD | RLN | Adv. | goggle | 0.678 | 0.320 | 0.678 | 0.000 | 0.005 | 0.000 |
| LCLD | RLN | Adv. | tablegan | 0.688 | 0.479 | 0.688 | 0.000 | 0.004 | 0.000 |
| LCLD | RLN | Adv. | tvae | 0.670 | 0.643 | 0.670 | 0.000 | 0.000 | 0.000 |

| | | | | | | | | | |
|---|---|---|---|---|---|---|---|---|---|
| LCLD | RLN | Adv. | wgan | 0.661 | 0.402 | 0.661 | 0.000 | 0.004 | 0.000 |
| LCLD | RLN | Std. | None | 0.683 | 0.000 | 0.683 | 0.000 | 0.000 | 0.000 |
| LCLD | RLN | Std. | ctgan | 0.705 | 0.001 | 0.705 | 0.000 | 0.001 | 0.000 |
| LCLD | RLN | Std. | cutmix | 0.689 | 0.000 | 0.689 | 0.000 | 0.000 | 0.000 |
| LCLD | RLN | Std. | goggle | 0.673 | 0.000 | 0.673 | 0.000 | 0.000 | 0.000 |
| LCLD | RLN | Std. | tablegan | 0.693 | 0.001 | 0.693 | 0.000 | 0.001 | 0.000 |
| LCLD | RLN | Std. | tvae | 0.700 | 0.000 | 0.700 | 0.000 | 0.000 | 0.000 |
| LCLD | RLN | Std. | wgan | 0.679 | 0.005 | 0.679 | 0.000 | 0.002 | 0.000 |
| LCLD | VIME | Adv. | None | 0.655 | 0.104 | 0.655 | 0.000 | 0.002 | 0.000 |
| LCLD | VIME | Adv. | ctgan | 0.789 | 0.768 | 0.789 | 0.000 | 0.000 | 0.000 |
| LCLD | VIME | Adv. | cutmix | 0.570 | 0.529 | 0.570 | 0.000 | 0.001 | 0.000 |
| LCLD | VIME | Adv. | goggle | 0.568 | 0.532 | 0.568 | 0.000 | 0.002 | 0.000 |
| LCLD | VIME | Adv. | tablegan | 0.563 | 0.537 | 0.563 | 0.000 | 0.000 | 0.000 |
| LCLD | VIME | Adv. | tvae | 0.678 | 0.661 | 0.678 | 0.000 | 0.001 | 0.000 |
| LCLD | VIME | Adv. | wgan | 0.617 | 0.530 | 0.617 | 0.000 | 0.002 | 0.000 |
| LCLD | VIME | Std. | None | 0.670 | 0.024 | 0.670 | 0.000 | 0.001 | 0.000 |
| LCLD | VIME | Std. | ctgan | 0.773 | 0.018 | 0.773 | 0.000 | 0.002 | 0.000 |
| LCLD | VIME | Std. | cutmix | 0.523 | 0.020 | 0.523 | 0.000 | 0.001 | 0.000 |
| LCLD | VIME | Std. | goggle | 0.644 | 0.005 | 0.644 | 0.000 | 0.001 | 0.000 |
| LCLD | VIME | Std. | tablegan | 0.607 | 0.005 | 0.607 | 0.000 | 0.001 | 0.000 |
| LCLD | VIME | Std. | tvae | 0.668 | 0.007 | 0.668 | 0.000 | 0.001 | 0.000 |
| LCLD | VIME | Std. | wgan | 0.659 | 0.007 | 0.659 | 0.000 | 0.002 | 0.000 |
| URL | STG | Adv. | None | 0.943 | 0.900 | 0.903 | 0.000 | 0.001 | 0.001 |
| URL | STG | Adv. | ctgan | 0.939 | 0.798 | 0.803 | 0.000 | 0.012 | 0.014 |
| URL | STG | Adv. | cutmix | 0.755 | 0.427 | 0.422 | 0.000 | 0.032 | 0.032 |
| URL | STG | Adv. | goggle | 0.939 | 0.856 | 0.860 | 0.000 | 0.010 | 0.008 |
| URL | STG | Adv. | tablegan | 0.921 | 0.809 | 0.816 | 0.000 | 0.004 | 0.003 |
| URL | STG | Adv. | tvae | 0.957 | 0.795 | 0.804 | 0.000 | 0.017 | 0.015 |
| URL | STG | Adv. | wgan | 0.942 | 0.812 | 0.813 | 0.000 | 0.003 | 0.003 |
| URL | STG | Std. | None | 0.933 | 0.580 | 0.596 | 0.000 | 0.008 | 0.007 |
| URL | STG | Std. | ctgan | 0.922 | 0.693 | 0.770 | 0.000 | 0.008 | 0.006 |
| URL | STG | Std. | cutmix | 0.794 | 0.397 | 0.444 | 0.000 | 0.009 | 0.010 |
| URL | STG | Std. | goggle | 0.939 | 0.745 | 0.759 | 0.000 | 0.005 | 0.006 |
| URL | STG | Std. | tablegan | 0.876 | 0.469 | 0.575 | 0.000 | 0.005 | 0.008 |
| URL | STG | Std. | tvae | 0.941 | 0.688 | 0.733 | 0.000 | 0.002 | 0.006 |
| URL | STG | Std. | wgan | 0.925 | 0.655 | 0.752 | 0.000 | 0.007 | 0.006 |
| URL | TabNet | Adv. | None | 0.995 | 0.918 | 0.919 | 0.000 | 0.002 | 0.001 |
| URL | TabNet | Adv. | ctgan | 0.901 | 0.899 | 0.899 | 0.000 | 0.000 | 0.000 |
| URL | TabNet | Adv. | cutmix | 0.930 | 0.897 | 0.896 | 0.000 | 0.001 | 0.001 |
| URL | TabNet | Adv. | goggle | 0.848 | 0.665 | 0.666 | 0.000 | 0.022 | 0.019 |
| URL | TabNet | Adv. | tablegan | 0.008 | 0.000 | 0.000 | 0.000 | 0.000 | 0.000 |
| URL | TabNet | Adv. | tvae | 0.940 | 0.872 | 0.870 | 0.000 | 0.018 | 0.018 |
| URL | TabNet | Adv. | wgan | 0.898 | 0.896 | 0.896 | 0.000 | 0.000 | 0.000 |
| URL | TabNet | Std. | None | 0.934 | 0.110 | 0.299 | 0.000 | 0.005 | 0.004 |
| URL | TabNet | Std. | ctgan | 0.994 | 0.948 | 0.948 | 0.000 | 0.002 | 0.001 |
| URL | TabNet | Std. | cutmix | 0.954 | 0.893 | 0.894 | 0.000 | 0.001 | 0.001 |
| URL | TabNet | Std. | goggle | 0.932 | 0.896 | 0.896 | 0.000 | 0.001 | 0.000 |
| URL | TabNet | Std. | tablegan | 0.896 | 0.878 | 0.875 | 0.000 | 0.010 | 0.011 |
| URL | TabNet | Std. | tvae | 0.938 | 0.891 | 0.892 | 0.000 | 0.002 | 0.003 |
| URL | TabNet | Std. | wgan | 0.998 | 0.952 | 0.953 | 0.000 | 0.002 | 0.001 |
| URL | TabTr | Adv. | None | 0.939 | 0.567 | 0.578 | 0.000 | 0.009 | 0.009 |
| URL | TabTr | Adv. | ctgan | 0.930 | 0.660 | 0.664 | 0.000 | 0.004 | 0.004 |
| URL | TabTr | Adv. | cutmix | 0.850 | 0.403 | 0.404 | 0.000 | 0.011 | 0.012 |
| URL | TabTr | Adv. | goggle | 0.917 | 0.541 | 0.554 | 0.000 | 0.006 | 0.007 |
| URL | TabTr | Adv. | tablegan | 0.898 | 0.409 | 0.421 | 0.000 | 0.010 | 0.011 |
| URL | TabTr | Adv. | tvae | 0.934 | 0.612 | 0.615 | 0.000 | 0.008 | 0.003 |
| URL | TabTr | Adv. | wgan | 0.927 | 0.569 | 0.580 | 0.000 | 0.008 | 0.010 |
| URL | TabTr | Std. | None | 0.936 | 0.089 | 0.825 | 0.000 | 0.002 | 0.001 |
| URL | TabTr | Std. | ctgan | 0.942 | 0.253 | 0.880 | 0.000 | 0.006 | 0.005 |

| | | | | | | | | | |
|---|---|---|---|---|---|---|---|---|---|
| URL | TabTr | Std. | cutmix | 0.904 | 0.018 | 0.687 | 0.000 | 0.000 | 0.000 |
| URL | TabTr | Std. | goggle | 0.930 | 0.049 | 0.051 | 0.000 | 0.001 | 0.001 |
| URL | TabTr | Std. | tablegan | 0.899 | 0.020 | 0.020 | 0.000 | 0.000 | 0.000 |
| URL | TabTr | Std. | tvae | 0.952 | 0.168 | 0.901 | 0.000 | 0.002 | 0.002 |
| URL | TabTr | Std. | wgan | 0.936 | 0.200 | 0.887 | 0.000 | 0.006 | 0.002 |
| URL | RLN | Adv. | None | 0.952 | 0.562 | 0.566 | 0.000 | 0.007 | 0.006 |
| URL | RLN | Adv. | ctgan | 0.938 | 0.625 | 0.628 | 0.000 | 0.005 | 0.007 |
| URL | RLN | Adv. | cutmix | 0.943 | 0.608 | 0.609 | 0.000 | 0.003 | 0.007 |
| URL | RLN | Adv. | goggle | 0.939 | 0.661 | 0.665 | 0.000 | 0.008 | 0.006 |
| URL | RLN | Adv. | tablegan | 0.913 | 0.555 | 0.557 | 0.000 | 0.009 | 0.005 |
| URL | RLN | Adv. | tvae | 0.941 | 0.598 | 0.602 | 0.000 | 0.003 | 0.003 |
| URL | RLN | Adv. | wgan | 0.933 | 0.547 | 0.552 | 0.000 | 0.002 | 0.005 |
| URL | RLN | Std. | None | 0.944 | 0.108 | 0.901 | 0.000 | 0.002 | 0.001 |
| URL | RLN | Std. | ctgan | 0.942 | 0.219 | 0.855 | 0.000 | 0.005 | 0.001 |
| URL | RLN | Std. | cutmix | 0.941 | 0.086 | 0.926 | 0.000 | 0.002 | 0.002 |
| URL | RLN | Std. | goggle | 0.936 | 0.039 | 0.039 | 0.000 | 0.000 | 0.000 |
| URL | RLN | Std. | tablegan | 0.910 | 0.039 | 0.039 | 0.000 | 0.000 | 0.000 |
| URL | RLN | Std. | tvae | 0.942 | 0.081 | 0.912 | 0.000 | 0.002 | 0.002 |
| URL | RLN | Std. | wgan | 0.935 | 0.214 | 0.911 | 0.000 | 0.002 | 0.002 |
| URL | VIME | Adv. | None | 0.934 | 0.698 | 0.727 | 0.000 | 0.006 | 0.004 |
| URL | VIME | Adv. | ctgan | 0.910 | 0.669 | 0.690 | 0.000 | 0.005 | 0.007 |
| URL | VIME | Adv. | cutmix | 0.920 | 0.686 | 0.707 | 0.000 | 0.010 | 0.012 |
| URL | VIME | Adv. | goggle | 0.919 | 0.737 | 0.749 | 0.000 | 0.013 | 0.011 |
| URL | VIME | Adv. | tablegan | 0.887 | 0.645 | 0.652 | 0.000 | 0.005 | 0.004 |
| URL | VIME | Adv. | tvae | 0.899 | 0.636 | 0.711 | 0.000 | 0.004 | 0.004 |
| URL | VIME | Adv. | wgan | 0.897 | 0.650 | 0.705 | 0.000 | 0.004 | 0.004 |
| URL | VIME | Std. | None | 0.925 | 0.495 | 0.533 | 0.000 | 0.005 | 0.003 |
| URL | VIME | Std. | ctgan | 0.927 | 0.548 | 0.910 | 0.000 | 0.004 | 0.001 |
| URL | VIME | Std. | cutmix | 0.925 | 0.467 | 0.913 | 0.000 | 0.004 | 0.001 |
| URL | VIME | Std. | goggle | 0.893 | 0.445 | 0.857 | 0.000 | 0.003 | 0.001 |
| URL | VIME | Std. | tablegan | 0.875 | 0.430 | 0.750 | 0.000 | 0.005 | 0.003 |
| URL | VIME | Std. | tvae | 0.909 | 0.444 | 0.886 | 0.000 | 0.005 | 0.003 |
| URL | VIME | Std. | wgan | 0.922 | 0.519 | 0.905 | 0.000 | 0.008 | 0.003 |
| WIDS | STG | Adv. | None | 0.626 | 0.452 | 0.626 | 0.000 | 0.002 | 0.000 |
| WIDS | STG | Adv. | ctgan | 0.853 | 0.738 | 0.853 | 0.000 | 0.002 | 0.000 |
| WIDS | STG | Adv. | cutmix | 0.532 | 0.412 | 0.523 | 0.000 | 0.001 | 0.003 |
| WIDS | STG | Adv. | goggle | 0.788 | 0.660 | 0.788 | 0.000 | 0.002 | 0.000 |
| WIDS | STG | Adv. | tablegan | 0.689 | 0.566 | 0.688 | 0.000 | 0.003 | 0.001 |
| WIDS | STG | Adv. | tvae | 0.677 | 0.598 | 0.677 | 0.000 | 0.001 | 0.001 |
| WIDS | STG | Adv. | wgan | 0.626 | 0.464 | 0.623 | 0.000 | 0.002 | 0.001 |
| WIDS | STG | Std. | None | 0.776 | 0.638 | 0.773 | 0.000 | 0.002 | 0.000 |
| WIDS | STG | Std. | ctgan | 0.878 | 0.712 | 0.877 | 0.000 | 0.003 | 0.000 |
| WIDS | STG | Std. | cutmix | 0.567 | 0.385 | 0.559 | 0.000 | 0.004 | 0.000 |
| WIDS | STG | Std. | goggle | 0.742 | 0.572 | 0.739 | 0.000 | 0.003 | 0.000 |
| WIDS | STG | Std. | tablegan | 0.677 | 0.498 | 0.671 | 0.000 | 0.004 | 0.000 |
| WIDS | STG | Std. | tvae | 0.759 | 0.621 | 0.755 | 0.000 | 0.003 | 0.000 |
| WIDS | STG | Std. | wgan | 0.746 | 0.583 | 0.744 | 0.000 | 0.002 | 0.000 |
| WIDS | TabNet | Adv. | None | 0.984 | 0.584 | 0.825 | 0.000 | 0.002 | 0.000 |
| WIDS | TabNet | Adv. | ctgan | 1.000 | 1.000 | 1.000 | 0.000 | 0.000 | 0.000 |
| WIDS | TabNet | Adv. | cutmix | 1.000 | 0.374 | 0.671 | 0.000 | 0.003 | 0.007 |
| WIDS | TabNet | Adv. | goggle | 1.000 | 1.000 | 1.000 | 0.000 | 0.000 | 0.000 |
| WIDS | TabNet | Adv. | tablegan | 1.000 | 1.000 | 1.000 | 0.000 | 0.000 | 0.000 |
| WIDS | TabNet | Adv. | tvae | 1.000 | 1.000 | 1.000 | 0.000 | 0.000 | 0.000 |
| WIDS | TabNet | Adv. | wgan | 1.000 | 0.992 | 0.996 | 0.000 | 0.004 | 0.002 |
| WIDS | TabNet | Std. | None | 0.797 | 0.053 | 0.731 | 0.000 | 0.004 | 0.002 |
| WIDS | TabNet | Std. | ctgan | 1.000 | 1.000 | 1.000 | 0.000 | 0.000 | 0.000 |
| WIDS | TabNet | Std. | cutmix | 0.000 | 0.000 | 0.000 | 0.000 | 0.000 | 0.000 |
| WIDS | TabNet | Std. | goggle | 1.000 | 1.000 | 1.000 | 0.000 | 0.000 | 0.000 |
| WIDS | TabNet | Std. | tablegan | 1.000 | 1.000 | 1.000 | 0.000 | 0.000 | 0.000 |

| WIDS | TabNet | Std. | tvae | 0.984 | 0.406 | 0.475 | 0.000 | 0.001 | 0.000 |
|------|--------|------|------|-------|-------|-------|-------|-------|-------|
| WIDS | TabNet | Std. | wgan | 0.786 | 0.000 | 0.456 | 0.000 | 0.000 | 0.003 |
| WIDS | TabTr | Adv. | None | 0.773 | 0.651 | 0.767 | 0.000 | 0.002 | 0.000 |
| WIDS | TabTr | Adv. | ctgan | 0.781 | 0.681 | 0.776 | 0.000 | 0.003 | 0.001 |
| WIDS | TabTr | Adv. | cutmix | 0.600 | 0.508 | 0.599 | 0.000 | 0.003 | 0.001 |
| WIDS | TabTr | Adv. | goggle | 0.765 | 0.675 | 0.755 | 0.000 | 0.002 | 0.001 |
| WIDS | TabTr | Adv. | tablegan | 0.726 | 0.622 | 0.724 | 0.000 | 0.002 | 0.001 |
| WIDS | TabTr | Adv. | tvae | 0.747 | 0.667 | 0.743 | 0.000 | 0.002 | 0.001 |
| WIDS | TabTr | Adv. | wgan | 0.765 | 0.652 | 0.759 | 0.000 | 0.002 | 0.002 |
| WIDS | TabTr | Std. | None | 0.755 | 0.459 | 0.746 | 0.000 | 0.003 | 0.000 |
| WIDS | TabTr | Std. | ctgan | 0.780 | 0.441 | 0.776 | 0.000 | 0.005 | 0.000 |
| WIDS | TabTr | Std. | cutmix | 0.710 | 0.434 | 0.705 | 0.000 | 0.003 | 0.000 |
| WIDS | TabTr | Std. | goggle | 0.786 | 0.383 | 0.733 | 0.000 | 0.004 | 0.000 |
| WIDS | TabTr | Std. | tablegan | 0.750 | 0.376 | 0.750 | 0.000 | 0.008 | 0.000 |
| WIDS | TabTr | Std. | tvae | 0.776 | 0.493 | 0.763 | 0.000 | 0.003 | 0.001 |
| WIDS | TabTr | Std. | wgan | 0.763 | 0.376 | 0.763 | 0.000 | 0.005 | 0.000 |
| WIDS | RLN | Adv. | None | 0.780 | 0.666 | 0.773 | 0.000 | 0.002 | 0.000 |
| WIDS | RLN | Adv. | ctgan | 1.000 | 1.000 | 1.000 | 0.000 | 0.000 | 0.000 |
| WIDS | RLN | Adv. | cutmix | 0.681 | 0.599 | 0.675 | 0.000 | 0.002 | 0.001 |
| WIDS | RLN | Adv. | goggle | 0.783 | 0.691 | 0.774 | 0.000 | 0.003 | 0.001 |
| WIDS | RLN | Adv. | tablegan | 0.760 | 0.661 | 0.754 | 0.000 | 0.002 | 0.001 |
| WIDS | RLN | Adv. | tvae | 0.775 | 0.711 | 0.772 | 0.000 | 0.003 | 0.002 |
| WIDS | RLN | Adv. | wgan | 0.776 | 0.676 | 0.776 | 0.000 | 0.003 | 0.001 |
| WIDS | RLN | Std. | None | 0.775 | 0.609 | 0.771 | 0.000 | 0.002 | 0.000 |
| WIDS | RLN | Std. | ctgan | 0.762 | 0.472 | 0.759 | 0.000 | 0.007 | 0.000 |
| WIDS | RLN | Std. | cutmix | 0.770 | 0.587 | 0.767 | 0.000 | 0.002 | 0.000 |
| WIDS | RLN | Std. | goggle | 0.773 | 0.525 | 0.750 | 0.000 | 0.001 | 0.000 |
| WIDS | RLN | Std. | tablegan | 0.788 | 0.589 | 0.786 | 0.000 | 0.004 | 0.000 |
| WIDS | RLN | Std. | tvae | 0.802 | 0.621 | 0.796 | 0.000 | 0.004 | 0.000 |
| WIDS | RLN | Std. | wgan | 0.775 | 0.574 | 0.775 | 0.000 | 0.002 | 0.000 |
| WIDS | VIME | Adv. | None | 0.721 | 0.521 | 0.721 | 0.000 | 0.003 | 0.000 |
| WIDS | VIME | Adv. | ctgan | 1.000 | 1.000 | 1.000 | 0.000 | 0.000 | 0.000 |
| WIDS | VIME | Adv. | cutmix | 0.543 | 0.435 | 0.535 | 0.000 | 0.002 | 0.001 |
| WIDS | VIME | Adv. | goggle | 0.702 | 0.592 | 0.699 | 0.000 | 0.002 | 0.001 |
| WIDS | VIME | Adv. | tablegan | 0.553 | 0.423 | 0.553 | 0.000 | 0.002 | 0.000 |
| WIDS | VIME | Adv. | tvae | 0.721 | 0.618 | 0.721 | 0.000 | 0.001 | 0.000 |
| WIDS | VIME | Adv. | wgan | 0.715 | 0.606 | 0.715 | 0.000 | 0.002 | 0.000 |
| WIDS | VIME | Std. | None | 0.723 | 0.503 | 0.713 | 0.000 | 0.002 | 0.000 |
| WIDS | VIME | Std. | ctgan | 1.000 | 1.000 | 1.000 | 0.000 | 0.000 | 0.000 |
| WIDS | VIME | Std. | cutmix | 0.699 | 0.476 | 0.694 | 0.000 | 0.002 | 0.000 |
| WIDS | VIME | Std. | goggle | 0.702 | 0.491 | 0.697 | 0.000 | 0.003 | 0.000 |
| WIDS | VIME | Std. | tablegan | 0.718 | 0.501 | 0.718 | 0.000 | 0.004 | 0.000 |
| WIDS | VIME | Std. | tvae | 0.726 | 0.506 | 0.726 | 0.000 | 0.004 | 0.000 |
| WIDS | VIME | Std. | wgan | 0.755 | 0.512 | 0.754 | 0.000 | 0.001 | 0.000 |

## B.4 Correlations between ID and robust performances

Table 11: Pearson correlations between constrained robust accuracy and: ID accuracy (ID), and non constrained-accuracy (ADV)

| Dataset | Training | ID(corr) | ID(p-val) | ADV(corr) | ADV(p-val) |
|---------|----------|----------|-----------|-----------|------------|
| CTU | Adversarial | 1 | 1.4e-26 | 1 | 1.9e-31 |
| CTU | Standard | 0.22 | 0.28 | 0.22 | 0.28 |
| LCLD | Adversarial | 0.76 | 1.8e-06 | 0.76 | 1.8e-06 |
| LCLD | Standard | 0.15 | 0.39 | 0.15 | 0.39 |
| URL | Adversarial | 0.7 | 3.6e-06 | 1 | 7.2e-37 |
| URL | Standard | 0.19 | 0.26 | 0.46 | 0.0053 |
| WIDS | Adversarial | 0.79 | 1e-06 | 0.91 | 7e-11 |
| WIDS | Standard | 0.031 | 0.87 | 0.62 | 0.00025 |

## B.5 Impact of budgets, detailed results

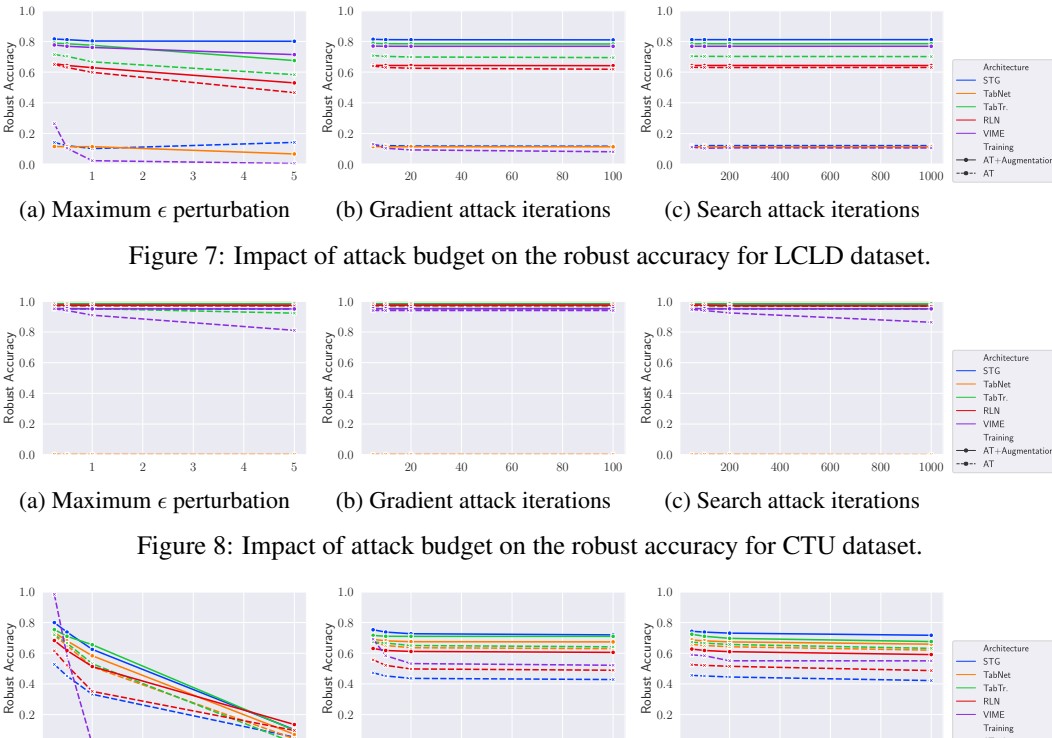

Figure 7: Impact of attack budget on the robust accuracy for LCLD dataset.

Figure 8: Impact of attack budget on the robust accuracy for CTU dataset.

Figure 9: Impact of attack budget on the robust accuracy for WIDS dataset.

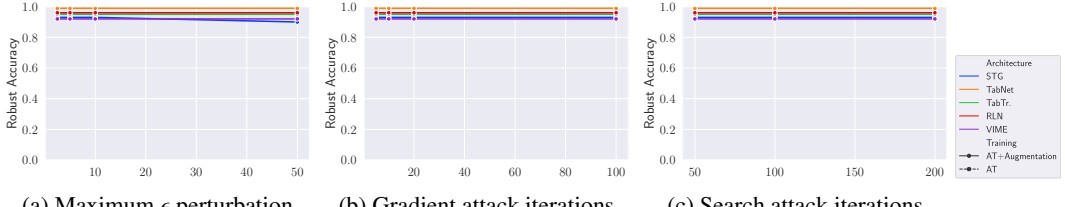

(a) Maximum $\epsilon$ perturbation     (b) Gradient attack iterations     (c) Search attack iterations

Figure 10: Impact of attack budget on the robust accuracy for Malware dataset.

## B.6 Generalization to other distances

We define for all attacks a distance function. This method is used for MOEVA (the evolution attack) to measure the fitness value related to the distance objective, and in the evaluation method to validate the correctness of the adversarial examples.

By default, it supports $L_\infty$ and $L_2$ distances [3]:

```python
from tabularbench.utils.typing import NDBool, NDInt, NDNumber

def compute_distance(x_1: NDNumber, x_2: NDNumber, norm: Any) ->
                                    NDNumber:
    if norm in ["inf", np.inf, "Linf", "linf"]:
        distance = np.linalg.norm(x_1 - x_2, ord=np.inf, axis=-1)
    elif norm in ["2", 2, "L2", "l2"]:
        distance = np.linalg.norm(x_1 - x_2, ord=2, axis=-1)
    else:
        raise NotImplementedError

    return distance
```

One can define any new distance metric, like structural similarity index measure (SSIM), or some semantic measure after embedding the features $x_1$ and $x_2$. The distance used here does not need to be differentiable and is not backpropagated in the gradient attacks.

Hence, for CAPGD component of the benchmark attack, we need to define a custom project mechanism for each distance. We implemented a projection over sphere of $L_\infty$ and $L_2$ distances `https://github.com/serval-uni-lu/tabularbench/blob/main/tabularbench/attacks/capgd/capgd.py#L196`.

To extend the projected gradient attacks to other distances, custom projection mechanisms are then needed.

# C  API

The library `https://github.com/serval-uni-lu/tabularbench/tree/main/tabularbench` is split in 4 main components. The *test* folder provides meaningful examples for each component.

## C.1  Datasets

Our dataset factory support 5 datasets: CTU, LCLD, MALWARE, URL, and WIDS. each dataset can be invoked with the following aliases:

```python
from tabularbench.datasets import dataset_factory

dataset_aliases= [
        "ctu_13_neris",
        "lcld_time",
        "malware",
        "url",
        "wids",
    ]

for dataset_name in dataset_aliases:
    dataset = dataset_factory.get_dataset(dataset_name)
    x, _ = dataset.get_x_y()
    metadata = dataset.get_metadata(only_x=True)
    assert x.shape[1] == metadata.shape[0]
```

---

[3] `https://github.com/serval-uni-lu/tabularbench/blob/main/tabularbench/attacks/utils.py`

Each dataset can be defined in a single .py file (example: `https://github.com/serval-uni-lu/tabularbench/blob/main/tabularbench/datasets/samples/url.py`).

A dataset needs at least a source (local or remote csv) for the raw features, and a definition of feature constraints. The said definition can be empty for non-constrained datasets.

## C.2 Constraints

One of the features of our benchmark is the support of feature constraints, in the dataset definition and in the attacks.

Constraints can be expressed in natural language. For example, we express the constraint $F_0 = F_1 + F_2$ such as:

```
from tabularbench.constraints.relation_constraint import Feature
constraint1 = Feature(0) == Feature(1) + Feature(2)
```

Given a dataset, one can check the constraint satisfaction over all constraints, given a tolerance.

```
from tabularbench.constraints.constraints_checker import
                                    ConstraintChecker
from tabularbench.datasets import dataset_factory

dataset = dataset_factory.get_dataset("url")
x, _ = dataset.get_x_y()

constraints_checker = ConstraintChecker(
    dataset.get_constraints(), tolerance
)
out = constraints_checker.check_constraints(x.to_numpy())
```

In the provided datasets, all constraints are satisfied. During the attack, Constraints can be fixed as follows:

```
import numpy as np
from tabularbench.constraints.constraints_fixer import
                                    ConstraintsFixer

x = np.arange(9).reshape(3, 3)

constraints_fixer = ConstraintsFixer(
        guard_constraints=[constraint1],
        fix_constraints=[constraint1],
    )

x_fixed = constraints_fixer.fix(x)

x_expected = np.array([[3, 1, 2], [9, 4, 5], [15, 7, 8]])

assert np.equal(x_fixed, x_expected).all()
```

Constraint violations can be translated into losses and one can compute the gradient to repair the faulty constraints as follows:

```
import torch

from tabularbench.constraints.constraints_backend_executor import (
    ConstraintsExecutor,
)

from tabularbench.constraints.pytorch_backend import PytorchBackend
from tabularbench.datasets.dataset_factory import get_dataset

ds = get_dataset("url")
```

```
constraints = ds.get_constraints()
constraint1 = constraints.relation_constraints[0]

x, y = ds.get_x_y()
x_metadata = ds.get_metadata(only_x=True)
x = torch.tensor(x.values, dtype=torch.float32)

constraints_executor = ConstraintsExecutor(
    constraint1,
    PytorchBackend(),
    feature_names=x_metadata["feature"].to_list(),
)

x.requires_grad = True
loss = constraints_executor.execute(x)
grad = torch.autograd.grad(
    loss.sum(),
    x,
)[0]
```

## C.3  Models

All models need to extend the class **BaseModelTorch**[4] . This class implements the definitions, the fit and evaluation methods, and the save and loading methods. Depending on the architecture, scaler and feature encoders can be required by the constructors.

So far, our API natively supports: multi-layer perceptrons (MLP), RLN, STG, TabNet, TabTransformer, and VIME. Our implementation is based on Tabsurvey Borisov et al. (2021). All models from this framework can be easily adapted to our API.

## C.4  Benchmark

The leaderboard is available on `https://serval-uni-lu.github.io/tabularbench/`.

This leaderboard will be updated regularly, and all the models listed in leaderboard are downloadable using our API

## TabularBench

TabularBench: Adversarial robustness benchmark for tabular data

**Leaderboard**

**CTU**                                                                    Search: STG

| architecture | training | augmentation | ID | ADV+CTR | ADV | auc | accuracy | precision | recall | mcc |
|---|---|---|---|---|---|---|---|---|---|---|
| STG | adversarial | tvae | 0.982801 | 0.982801 | 0.98231 | 0.981094 | 0.435641 | 0.0127069 | 0.982801 | 0.0717109 |
| STG | standard | tvae | 0.963145 | 0.960688 | 0.963145 | 0.984115 | 0.890109 | 0.0609642 | 0.963145 | 0.227425 |
| STG | adversarial | ctgan | 0.960688 | 0.960197 | 0.959214 | 0.986319 | 0.929578 | 0.0919135 | 0.960688 | 0.285528 |
| STG | adversarial | wgan | 0.953317 | 0.953317 | 0.953317 | 0.984742 | 0.999528 | 0.982278 | 0.953317 | 0.967453 |
| STG | standard | None | 0.953317 | 0.953317 | 0.953317 | 0.988398 | 0.999528 | 0.982278 | 0.953317 | 0.967453 |
| STG | standard | ctgan | 0.955774 | 0.953317 | 0.955774 | 0.990381 | 0.998802 | 0.89016 | 0.955774 | 0.92179 |
| STG | standard | cutmix | 0.953317 | 0.953317 | 0.953317 | 0.986111 | 0.999528 | 0.982278 | 0.953317 | 0.967453 |

Figure 11: Screenshot of the TabularBench leaderboard on 12/06/2024

---

[4]`https://github.com/serval-uni-lu/tabularbench/blob/main/tabularbench/models/torch_models.py`

The benchmark leverages Constrained Adaptive Attack (CAA) by default and can be extended for other attacks.

```
clean_acc, robust_acc = benchmark(dataset='LCLD', model="TabTr_Cutmix"
                                  , distance='L2', constraints=True)
```

The model attribute refers to a pre-trained model in the relevant model folder. The API infers the architecture from the first term of the model name, but it can be defined manually. In the above example, a **TabTransformer** architecture will be initialized.

