# OpenReview forum: "TabularBench: Benchmarking Adversarial Robustness for Tabular Deep Learning in Real-world Use-cases"
_NeurIPS.cc/2024/Datasets_and_Benchmarks_Track — NeurIPS 2024 Track Datasets and Benchmarks Poster_

### Official Review · Reviewer_cirj · 2024-07-13
**Review comments for TabularBench**

**Rating:** 5
**Confidence:** 3

**Review:**

The TabularBench submission demonstrates high quality through its comprehensive evaluation of over 200 models across different domains, implementation of advanced attack mechanisms, and clear presentation of evaluation settings. The work is original in its pioneering initiative in the field of tabular adversarial attacks and its use of robustification techniques inspired by top-performing research in computer vision. The significance of TabularBench lies in its potential to advance the understanding of adversarial robustness in tabular deep learning models and its practical implications for real-world applications.

However, there are some areas that could be further addressed, such as the scalability of the benchmark to larger datasets or more complex models, detailed analysis on the computational resources required for running the benchmark, exploration of the transferability of defense mechanisms across different domains, generalization of findings to diverse tabular data scenarios, and discussion on the interpretability of adversarial attacks and defenses in tabular models. Addressing these areas could enhance the overall impact and applicability of the TabularBench benchmark in the field of adversarial robustness evaluation for tabular deep learning models.

**Strengths:**

The strengths of the TabularBench submission include its pioneering contribution as the first comprehensive benchmark for evaluating the robustness of tabular deep learning models in real-world scenarios. This work is highly relevant to the research community, setting a new standard for adversarial robustness evaluation in tabular data. The quality of the research is evident in its rigorous evaluation process and clear presentation of methodologies. Ethically, TabularBench promotes the development of more secure and trustworthy AI technologies, with implications for various real-world applications.

**Additional Feedback:**

While the excerpts mention the use of advanced attack and defense mechanisms, ensuring that the benchmark documentation includes detailed information on data collection, organization, availability, and maintenance plans would further support reproducibility and transparency.

It would be valuable for the authors to provide a more detailed comparison of TabularBench with existing benchmarks in terms of evaluation metrics, attack scenarios, and realism of assessments. This could help highlight the unique contributions and advantages of TabularBench in the field.

It could be beneficial for the authors to discuss potential future directions for expanding TabularBench, such as incorporating additional datasets, exploring different attack strategies, or extending the benchmark to address evolving challenges in adversarial robustness for tabular models.

**Clarity:**

Based on the excerpts provided from the TabularBench submission, the paper appears to be well-written. The language used is clear and concise, making it easy to understand the objectives, methodology, and contributions of the work. The authors effectively communicate the novelty of their benchmark for evaluating the robustness of tabular deep learning models in real-world use cases. Additionally, the paper seems to provide a structured and coherent presentation of the dataset construction process, evaluation methods, and experiment design.

**Correctness:**

Based on the information provided in the TabularBench submission, the claims made in the submission appear to be correct. The authors have highlighted the novelty of their work as the first comprehensive benchmark for evaluating the robustness of tabular deep learning models in real-world settings. They have also emphasized the importance of addressing the lack of standardized benchmarks in the field of adversarial robustness evaluation for tabular data.

**Documentation:**

The authors have curated datasets meeting specific criteria, such as being open source, from real-world applications, and containing feature relationships and constraints. This detailed information on dataset selection criteria and characteristics enhances the transparency and reproducibility of the benchmark.

Additionally, the submission mentions the use of the Constrained Adaptive Attack (CAA) and advanced defense mechanisms inspired by top-performing research in computer vision. By incorporating state-of-the-art attack and defense techniques, the authors have ensured that the benchmark evaluation methods are well-documented and designed to support reproducibility.

While the excerpts do not provide specific details on data collection and organization, availability and maintenance, or ethical and responsible use, the focus on dataset curation criteria and attack/defense mechanisms suggests that the benchmark documentation likely includes relevant information on these aspects. To fully assess the completeness of the documentation, reviewers may need to refer to the full submission or supplementary materials for detailed information on data collection, organization, availability, ethical considerations, and reproducibility support.

**Ethics:**

Based on the information provided in the excerpts from the TabularBench submission, there are no explicit indications of ethical concerns related to the dataset curation, benchmark evaluation, or research methodology. The authors mention that the datasets used in the benchmark are open source and publicly available, meeting specific criteria for inclusion in the evaluation of adversarial robustness for tabular deep learning models.

**Limitations:**

To further address these limitations and potential negative societal impacts, the authors could provide more detailed discussions and potential solutions:

Scalability: Given the complexity of tabular datasets and deep learning models, the authors could explore strategies for scaling the benchmark to larger datasets and more complex models. This could involve discussing techniques like data parallelism, model parallelism, or algorithmic optimizations to handle scalability issues effectively.

Computational Resources: Providing a detailed analysis of the computational resources required to run the benchmark would enhance reproducibility and transparency. Including information on hardware specifications, runtime considerations, and potential optimizations would help researchers assess the feasibility of implementing the benchmark in their own settings.

Transferability of Defense Mechanisms: To improve the transferability of defense mechanisms across different domains, the authors could conduct experiments on a diverse set of tabular datasets. By evaluating the performance of defense mechanisms in various contexts, they can provide insights into the robustness and adaptability of these techniques.

Generalizability of Findings: Expanding the evaluation to include a wider range of tabular datasets representing different domains and characteristics would strengthen the generalizability of the benchmark results. This approach would offer a more comprehensive understanding of the effectiveness of adversarial robustness mechanisms in diverse real-world scenarios.

Interpretability of Adversarial Attacks and Defenses: Providing insights into the interpretability of adversarial attacks and defenses in tabular models would enhance the understanding of model vulnerabilities and security implications. Discussing the trade-offs between interpretability and robustness can help researchers make informed decisions about model security.

**Opportunities For Improvement:**

The limitations of the TabularBench submission include potential challenges in scalability to larger datasets or more complex models, the need for detailed analysis on the computational resources required for running the benchmark, exploration of the transferability of defense mechanisms across different domains, generalization of findings to diverse tabular data scenarios, and discussion on the interpretability of adversarial attacks and defenses in tabular models. Addressing these areas could enhance the overall impact and applicability of the TabularBench benchmark in the field of adversarial robustness evaluation for tabular deep learning models.

**Relation To Prior Work:**

The TabularBench clearly discusses how their work differs from previous contributions in the field of adversarial robustness evaluation for tabular deep learning models. The authors highlight the lack of standardized benchmarks for assessing the robustness of tabular models, emphasizing the gap in research on evasion attacks against tabular deep learning and the limited investigation of robustification mechanisms and defenses in this domain.

Specifically, the authors point out that existing adversarial attacks designed for tabular data often overlook feature-type constraints or only consider categorical features without accounting for feature relationships. They also mention that previous benchmarks focused on computer vision and natural language processing tasks, lacking a dedicated benchmark for tabular model robustness. By introducing TabularBench as the first comprehensive benchmark for evaluating the adversarial robustness of tabular deep learning models, the authors clearly position their work as a novel contribution that addresses the shortcomings of existing research in this area.

**Summary And Contributions:**

The submission introduces TabularBench, the first comprehensive benchmark for evaluating the adversarial robustness of tabular deep learning models in real-world use cases. The key contributions of TabularBench include addressing the lack of standardized benchmarks in the field of tabular adversarial attacks, evaluating over 200 models across various domains, and providing a valuable resource for researchers and practitioners to assess the robustness of tabular deep learning models.

---

### Official Review · Reviewer_EokF · 2024-07-22
**initial review**

**Rating:** 6
**Confidence:** 4
**Correctness:** Correctness is bound to the correctne…

**Review:**

While the presented benchmark and initial analysis are suitably structured, well organised and clearly filling a gap in the current research on robustness of tabular data algorithms, the paper has a central flaw: it solely relies on the CCA attack (Simonetto et al., 2024). This causes several major problems:

1. CCA is never properly introduced, making the paper not self-contained
2. at the same time, CCA is an un-reviewed recent arxiv paper and as such not (yet) accepted by the community
3. following 1+2 would require the reviewer to review two paper at once in order check the validity of the proposed benchmark
4. CCA is from the same authors as the benchmark paper. This paper might appear to be blind, but the provided github repo is from the CCA authors - which is not a problem per se (as the data set track allows non blind submissions), but causes another problem -> 5
5. given the template, CCA is most likely under review in the NeurIPs main track. Accepting the a CCA benchmark could lead to the odd situation that a benchmark entirely relying on a method could be accepted while the method is rejected at the same time ...

I think that the authors are moving to fast, trying to make two steps at once. Either the method and benchmark should be one paper, or the benchmark should be delayed until the method is accepted.

**Strengths:**

The is certainly a need for a tabular robustness benchmark. Given that the CCA attack is indeed a valid method, the well organized TabularBench would be a very useful for the advancement of our field...

**Additional Feedback:**

I think that the authors are moving to fast, trying to make two steps at once. Either the method and benchmark should be one paper, or the benchmark should be delayed until the method is accepted.

If the organisational logistics allow this, I'm willing to accept this paper IFF PCs would notify me that the CCA paper will be accepted in the main track - otherwise I will argue strongly against it.

UPDATE score 3->6 after rebuttal and discussion

**Clarity:**

Paper is not self-containing. At least a shallow introduction of CCA should be given. Otherwise the paper is very clear.

**Documentation:**

Dataset documentation and accessibility are very good.

**Ethics:**

no ethics concerns

**Limitations:**

Limitations are addressed properly

**Opportunities For Improvement:**

NA

**Relation To Prior Work:**

Again, the relation to the simultaneous CCA paper is not clearly entangled. Otherwise, related work appears to be cited properly.

**Summary And Contributions:**

The paper presents TabularBench, a benchmark for the evaluation of adversarial robustness of binary tabular classification. Following the setup of the widely used "RobustBench" for image classification, TabularBench provides a dataset zoo, model zoo, public benchmark and an initial analysis of the robustness of common tabular classification methods.

---

> ### Author Rebuttal · Authors · 2024-08-16
>
> Thank you for your feedback and your praise for the significance of our work in advancing tabular adversarial machine learning. We appreciate the opportunity to clarify and address misunderstandings about our benchmark. We will address each of your points one by one and welcome further discussion on these issues.
>
> We understand that you do not have any specific criticism on the methodology, the content, or the evaluation of the benchmark, but your concern is about the soundness and relevance of the CAA attack we leverage to build our benchmark.
>
> While we agree that the CAA attack is not introduced in detail in our benchmark. However, it is thoroughly evaluated in [A] and [B]. Indeed, the article of CAA was not yet published in a peer-reviewed venue, but the first introduction of this attack backs to 2023 on simple datasets (URL, LCLD). To the best of our knowledge, the fact that an attack was only published on public arxiv does not constitute a reason to discard a benchmark paper.
>
> Next, we would like to emphasize that CAA is the combination of two attacks that were each peer-reviewed and presented in reputable venues. MOEVA [C] and CAPGD [D]. The works in [A] and [B] demonstrate the superiority of CAA over each MOEVA and CAPGD. Hence, we believe our benchmark is based on robust techniques. Moreover, our benchmark remains relevant and supports the use of CAA as a whole or only MOEVA or CAPGD individually.
>
> Thank you for checking the CAA paper [A]. It is indeed under review in the technical track of NeurIPS with good scores. If you have any criticism on the soundness of CAA based on the pre-print, we would be happy to discuss these, and we agree with you that demonstrated critical flaws of CAA could be grounded reasons to reject our benchmark.
>
> We appreciate your suggestion to accept the paper IFF the CAA paper is accepted, but we do not think this is possible for calendar and logistical reasons, given that both tracks follow the same rebuttal and notification dates. We will contact the chair to ask if we/they can reveal you the exact final scores of CAA reviews, under their control.
>
> [A] Simonetto, Thibault, et al. "Constrained Adaptive Attack: Effective Adversarial Attack Against Deep Neural Networks for Tabular Data." arXiv preprint arXiv:2406.00775 (2024).
>
> [B] Simonetto, Thibault, et al. "Constrained Adaptive Attacks: Realistic Evaluation of Adversarial Examples and Robust Training of Deep Neural Networks for Tabular Data." arXiv preprint arXiv:2311.04503 (2023).
>
> [C] Simonetto, Thibault, et al. "A Unified Framework for Adversarial Attack and Defense in Constrained Feature Space." IJCAI 2022
>
> [D] Simonetto, Thibault, et al. "Towards Adaptive Attacks on Constrained Tabular Machine Learning". ICML 2024 Next Generation of AI Safety Workshop.

---

> > ### Comment · Reviewer_EokF · 2024-08-17
> > **Next steps**
> >
> > Thank you for your open answer, confirming that CAA is indeed under review at the main track.
> >
> > I will engage in discussions with my fellow reviewers and the AC on how to handle this special case. Indeed, my concerns are more on the procedural side than technical and I'm motivated to actively work on solution.

---

> ### Author Response · Authors · 2024-08-19
>
> Thank you for your understanding and your engagement in continuing the discussion. We renew our willingness to share any legitimate information useful for this purpose.
>
> Following our discussion with reviewer ZSGk we would like to point out that, while our experiments indeed rely on CAA, our benchmark implementation is not restricted to this attack and can be extended with new attacks by ourselves or by other researchers. As suggested by reviewer #ZSGk, we have updated the repository and the documentation to clearly explain how our benchmark supports new attacks [(HERE)](https://serval-uni-lu.github.io/tabularbench/doc/attacks.html).
>
> Beyond the evaluation of CAA on larger settings, this submission also brings novelties that were not present in the CAA paper and are (to some extent) independent of the attack:
>
> - Implement and evaluate five new adversarial-training based data-augmentation techniques.
> - Demonstrate the mismatch between adversarial robustness with and without constraints, revealing the need for the community to investigate more carefully the adversarial robustness under constraints.
> - Provide a dataset zoo for constrained tabular deep learning, ranging from simple cases where no architecture is robust (URL), to extremely challenging cases where all architectures are robust (MALWARE). This zoo is a springboard to new attack mechanisms to correctly assess complex cases.
> - Set up simple grammar and factory for users to create easily new constrained tabular datasets and evaluate them with a large pool of attacks and architectures [(HERE)](https://serval-uni-lu.github.io/tabularbench/doc/constraints.html).
> - Set up a live public leaderboard, with pre-trained models accessible to the community and a simple API for a standardized definition and benchmarking of tabular models and attacks.
>
> All these new contributions are independent of CAA, and we believe the insights they bring on the SOTA architectures and defenses for tabular machine learning will significantly speed up the research on tabular ML robustness, and hopefully catch up to the maturity of adversarial robustness in Computer Vision. Our hope to speed up the research and support the small community of Tabular ML was one of the reasons for us to submit the benchmark and the library in parallel with the CAA submission.

---

> > ### Comment · Reviewer_EokF · 2024-08-20
> >
> > Thank you for the update. After some internal discussion, I'm pretty much in line with reviewer ZSGk. Opening the Benchmark for other method is definitely a big step in the right direction and I will not oppose acceptance. I will raise my score accordingly.
> >
> > However, I still think that the acceptance of the benchmark with be strongly tied to the acceptance of CAA and that it would proper scientific procedure to to establish the method first and the use it to establish a benchmark.

---

> > > ### Author Rebuttal · Authors · 2024-09-01
> > >
> > > Dear reviewer,
> > >
> > > Thank you very much for your insights and feedback to improve our manuscript, and for raising your score from 3 to 5.  We appreciate the open discussion and your suggestions and we would like to reassure you of our commitment to include them all in the final version.
> > >
> > > Regards,

---

### Official Review · Reviewer_ZSGk · 2024-07-25

**Rating:** 7
**Confidence:** 3

**Review:**

The paper makes an important contribution by introducing a new benchmark to evaluate the adversarial robustness of models trained on tabular data. The benchmark is well motivated and the authors provide a detailed explanation of how they developed it. The paper is well written and easy to follow. The authors also provide a Python library to run and evaluate other models/defenses on this benchmark, which is an important addition to make sure the benchmark results are easily kept up-to-date. The results are interesting and show that adversarial training is effective at improving robustness, but that it is not the only factor that matters. The paper is well structured and the authors provide a good discussion of their results. The code is publicly released and has a README that explains how to use it. The paper, however, still has some limitations: the documentation on the GitHub repository can be improved and the use of the library could be made easier by allowing users to `pip install` it. Moreover, the authors do not mention the possibility to add results from adaptive attacks to the leaderboard. This would very important as adaptive attacks are sometimes the only way to effectively evaluate the robustness of a model.

**Strengths:**

This work has several solid strengths:

- A benchmark for adversarial robustness on models trained for tabular data classification tasks is important
- The selection criteria of the datasets are well motivated
- The authors provide a Python library to run the benchmark on other models and defenses
- The results are somewhat unsurprising (AT is more robust than data augmentation-based defenses), but still interesting to show on a large set of models and defenses
- The paper is well written and easy to follow

**Additional Feedback:**

The authors mention that some features could are computed internally and added to the model input by the model owner. However, it is unclear to me how this affects an attack, as these features would act just like a constant. Can the authors please clarify this point?

**Clarity:**

The paper is well written and easy to follow. However, the last paragraph of section 2 has some typos: madry -> Madry and add a citation to the AT paper, and in the second-last sentence "Robustbench" is repeated.

**Correctness:**

The claims made in the submission are correct. The benchmark is well constructed and the evaluation methods are appropriate.

**Documentation:**

The documentation on the GitHub repository can be improved: the authors should add a license to the repository, add the docs to the README (even just by adding the details present in Appendix C), and add instructions to `pip install` the library. They should also add instructions on how to add new results to the leaderboard, ideally with an Issue Template on GitHub.

**Ethics:**

No.

**Limitations:**

The authors discuss the limitations of their work. However, they should also clarify that evaluating a model against a standard attacks might not always be enough, and encourage researchers to evaluate their models against adaptive attacks as well.

**Opportunities For Improvement:**

The paper has however some limitations:

- The paper does not mention the possibility to add results from adaptive attacks to the leaderboard. The authors should add this feature as adaptive attacks are sometimes the only way to effectively evaluate the robustness of a model.
- Code does not have a license (even if MIT license is mentioned in the Appendix, but it should be in the repository itself)
- Docs at the end of the appendix should also be in the README or on the website
- No instructions to `pip install` the library
- No instructions on how to add new results to the leaderboard

These are all easy to address, and I will be happy to increase my score if they indeed are.

**Relation To Prior Work:**

The authors clearly discuss how their work differs from previous contributions.

**Summary And Contributions:**

This paper introduces a new benchmark to evaluate the adversarial robustness of models trained on tabular data against evasion attacks. After discussing why such benchmark is needed, they explain how they developed it. They first select five datasets suitable for the benchmark and an attack to use as the standard way to measure robustness of models and defense techniques. Then, they evaluate several combinations of data augmentation techniques, architectures, and training procedures (with and without adversarial training) and study the results of their benchmark. Finally, they release an open-source Python library to run and evaluate other models/defenses on this benchmark.

---

> ### Author Rebuttal · Authors · 2024-08-16
>
> Thank you for your positive feedback and praise for the extensiveness and significance of our work in advancing tabular adversarial machine learning. We appreciate your interest and would like to clarify and answer your concerns below.
>
> **W1: Possibility to add results from adaptive attacks:**
>
> Thank you for this suggestion; it is now possible for any github user to submit new attacks using the Issue feature of our github repository. We have prepared a simple form to complete by the contributor and a clear process for validation and integration in the API.
>
> We have not designed specific adaptive attacks for each defense mechanism, given that CAA that we use in the benchmark is itself a multi-purpose adaptive attack. We acknowledge that specific new attacks can be more effective against a specific defense or artifacts.
>
> For example, STG [A] architecture uses stochastic gates to select the features on the flow in training and inference, which makes CAA attack less effective on this architecture. One could specifically design an attack for this architecture that takes into account the stochasticity.
>
> Our work follows the philosophy of Robustbench [B] from the computer vision community. To achieve a general purpose benchmark they: "rule out classifiers which have zero gradients with respect to the input, (2) randomized classifiers, and (3) classifiers that use an optimization loop at inference time".
>
> Similarly, we did not cover specific adaptive attacks against random mechanisms, but we welcome new contributions tailored to this particular niche of defenses and architectures.
>
> We will update the motivation of CAA as a standarized general-purpose benchmark attack, and extend the limitation section with a discussion on other mechanisms that could be covered with dedicated adaptive attacks.
>
>
> [A] Yamada, Yutaro, et al. "Feature selection using stochastic gates." ICML, 2020.
>
> [B] Croce et al. "Robustbench: a standardized adversarial robustness benchmark."(NeurIPS, 2021).
>
> **W2: Code does not have a license**
>
> Thank you for pointing out this missing file. Indeed, the project's code is under the MIT License (as explained in the appendix), but each dataset uses its own license (MIT, CC-BY, CC0). We have added a MIT license file to the benchmark github and referenced each model and attack its license.
>
> **W3: Docs at the end of the appendix should also be in the README or on the website**
>
> Thank you for this suggestion. We have updated the github repository with the link to a dedicated documentation. This extensive documentation provides examples and guides on how to run the benchmark, use the datasets and constraints, and a detailed API of the attacks.
> In dedicated sections, we have also explained how to build a new dataset, a new model, a new attack, and how to contribute these elements to the benchmark and to the public leaderboards.
>
> **W4: No instructions to PIP install the library**
>
> Thank you for this feedback. We have now published our benchmark and library on PIP. IT can be now installed using:
>
> ```bash
>     pip install tabularbench
> ```
>
> We have updated the README and documentation accordingly.
>
> **W5: Possibility to add new results to the leaderboard:**
>
> Thank you for this suggestion; we were working on mechanisms to submit new results and explored many options. Our final solution is in line with your suggestion. We have implemented on github templated issues with structured forms to submit a new dataset, a new model, or a new attack. We have also updated the documentation to explain how to submit a new element with step-by-step instructions.

---

> > ### Comment · Reviewer_ZSGk · 2024-08-19
> >
> > Thank you for your response and for addressing the issues I raised. I agree with reviewer EokF that the paper should include a description of CAA--at least in the appendix.
> >
> > Furthermore, I would like to argue that, despite the name, a fixed attack (such as CAA) cannot be **truly** adaptive (even if it behaves differently for different defenses). In the very example you made of a randomized defense, one could probably make the defense less effective by using Expectation over Transformation [1, 2], which a fixed attack might not use out-of-the-box. For example, AutoAttack has some _automatic_ component to it (e.g., varying step sizes in AutoPGD), but it is not an adaptive attack itself. Hence, I do believe that  the importance of adaptive attacks should be remarked in the paper. Moreover, while RobustBench is almost fully based on AutoAttack, the paper clearly states that it accepts adaptive evaluations, and authors implemented this feature with a "Best known robust accuracy" column in the [leaderboard](https://robustbench.github.io). These two accuracies can sometimes differ by a non-negligible amount (hence showing that even AutoAttack might not be enough, as much as it has automatic components to it). I strongly recommend that you follow a similar approach.
> >
> > Moreover, I recommend that the authors remain open to the adoption of other attacks to use as the benchmark's standard in case CAA does not get established as the go-to attack for the tabular ML community, or if a significantly better attack comes out in the future.
> >
> > I have one final question, that just came to my mind (and I could not find in the paper). How long does it take to run the benchmark with the hardware stated in Appendix A.2? I believe that it would be important to report an estimate in the paper, so that people who want to run the benchmark have an idea of how long it is going to take.
> >
> > I will wait for a reply to my concerns, as well as the discussion with reviewer EokF on the relationship to the CAA paper (which I am actively participating to) , before raising my score.
> >
> > FYI, I won't be able to reply and/or update my score between August 21st and August 29th, so in case I do not, it is not because I forgot about your submission. I will make sure to actively participate in the discussion before and after that :)
> >
> > References
> >
> > - [1] Athalye et al., https://arxiv.org/abs/1707.07397
> > - [2] Athalye et al., https://arxiv.org/abs/1802.00420

---

> ### Author Response · Authors · 2024-08-19
>
> Thank you for your feedback and your commitment to continue the discussion!
>
> With this benchmark paper, our end goal is to set out the foundations for boosting research on adversarial robustness for tabular ML models and datasets. Our endeavor will continue beyond the submission, e.g. as documented in the roadmap of the leaderboard documented in our repository. In this respect, we appreciate your suggestions (including. The "best known robust accuracy" metric) that will definitely feed our roadmap. Thank you for this.
>
> Since CAA is currently the main attack our benchmark relies on, we agree that it would be better to include it in the Appendix of the paper to be self-contained. However, we would like to point out that (i) our code is open for other researchers to incorporate new attacks (following your suggestion, we have updated our repository and documentation to precisely explain how to do so), (ii) similar to Robustbench, we also encourage the community to submit the best know robust accuracy on adaptive attacks when submitting a new model to the leaderboard ([HERE](https://github.com/serval-uni-lu/tabularbench/issues/new?assignees=&labels=contribution&projects=&template=model_submission.yml)) and (iii) this submission brings new insights beyond the CAA paper and the related literature:
>
> - Implement and evaluate six new adversarial-training based data-augmentation techniques.
> - Demonstrate the mismatch between adversarial robustness with and without constraints, revealing the need for the community to investigate more carefully the adversarial robustness under constraints.
> - Provide a dataset zoo for constrained tabular deep learning, ranging from simple cases where no architecture is robust (URL), to extremely challenging cases where all architectures are robust (MALWARE). This zoo is a springboard to new attack mechanisms to correctly assess complex cases.
> - Set up simple grammar and factory for users to create easily new constrained tabular datasets and evaluate them with a large pool of attacks and architectures ([HERE](https://serval-uni-lu.github.io/tabularbench/doc/constraints.html)).
> - Set up a live public leaderboard, with pre-trained models accessible to the community and a simple API for a standardized definition and benchmarking of tabular models and attacks.
>
> All these new contributions are independent of CAA, and we believe the insights they bring on the SOTA architectures and defenses for tabular machine learning will significantly speed up the research on tabular ML robustness, and hopefully catch up to the maturity of adversarial robustness in Computer Vision. These insights, combined with our roadmap and the fact that our code architecture can accept new attacks, give us hope that our benchmark can achieve such an impact.
>
> Regarding the time needed to run the benchmark using CAA, we provide below the execution time in seconds for each dataset and architecture. We run the benchmark on 1.000 examples each with the standard benchmark parameters (100 iterations, no time limit). Given the search component MOEVA within CAA, the execution time linearly increases with the complexity of the dataset. The malware dataset that we curated for this benchmark is very robust to CAA (CF Table 3 of our benchmark) and extremely costly to attack. These properties make it a suitable use case to evaluate future attacks with our benchmark.
>
> Thank you for raising this point, we will update the appendix of the manuscript with these results.
>
>
> |             | TabTransformer   | RLN   | VIME  | STG   | TabNet |
> |-------------|---------------|------------|------------|------------|-------------|
> | **URL**   | 17±1       | 19±1       | 51±2       | 73±3       | 58±1        |
> | **LCLD**   | 83±2       | 10±3       | 13±1       | 57±2       | 23±0        |
> | **WIDS**   | 49±1       | 49±2       | 41±1       | 59±1       | 25±1        |
> | **CTU**   | 110±5       | 112±4       | 116±2       | 119±4       | 182±4        |
> | **MALWARE**   | 1509±125       | 936±116 | 3006±47 | 971±103 | 4008±220 |

---

> > ### Author Response · Authors · 2024-08-30
> >
> > Dear reviewer,
> >
> > We understand you mentioned being away for a few days, and we wanted to check in regarding our last comment.
> >
> > We appreciate the time and effort you are putting into the review process and would be grateful if you could provide an update at your earliest convenience, as the discussion period is ending soon.
> >
> > Thank you for your consideration.

---

> ### Comment · Reviewer_ZSGk · 2024-08-31
>
> Thank you for your reply and for the details on the running time of the benchmark. Trusting that you will indeed add a description of CAA in the paper, I am happy to raise my score to 7. As much as the paper's contribution go beyond the use of CAA, the attack is also the core of the benchmark, which, in turn, is the core of the paper, hence, a description of the attack would be very useful (especially as the attack is a very recent contribution).
>
> Edit: turns out I can't edit my review for some reason. But be reassured that I will communicate my score change to the AC.

---

> > ### Author Rebuttal · Authors · 2024-09-01
> >
> > Dear reviewer,
> > thank you very much for your feedbacks and for increasing your score to 7.  We appreciate the discussion and insights your provided us and assure you all your suggestions will be included in the final version.
> > Regards,

---

### Author Rebuttal · Authors · 2024-08-16

We thank the reviewers for their comments. The reviewers agree on the importance of the problem we tackle and are satisfied with the comprehensiveness of our benchmark and analyses.

Our work proposes the first benchmark of constrained tabular attacks on deep tabular classification model.

Our benchmark leverages the CAA attack, the most effective and efficient attack for constrained tabular machine learning, and implements 7 SoTA defenses with adversarial training to evaluate CAA under challenging settings.

We distribute an exhaustive library with a dataset zoo over five real-world datasets and a model zoo comprised of 200 pretrained models over five architectures.

We have addressed the suggestions of the reviewers for a detailed documentation, distributed the benchmark and library in PIP, and implemented processes and documentation for the community to contribute new datasets, attacks, and robust models.

We hope these improvements answer the expectations of the reviewers. We are open to further discussion and would be happy to elaborate on any points if needed.

---

> ### Author Response · Authors · 2024-09-01
>
> Dear reviewers,
>
> Thank you for your valuable suggestions and for taking the time to engage with us during the discussion period.
>
> We will carefully consider your feedback for the camera-ready version of the paper if accepted.
>
> Specifically, we plan to use the additional page to include: 1) a detailed description of the CAA attack 2) its limitations w.r.t. adaptive attacks 3) how our benchmark can be used beyond the CAA attack.

---

> > ### Comment · Reviewer_EokF · 2024-09-01
> >
> > I think that the author's promises are genuine and realistic. Given these changes I'm ok with accepting the paper and will raise my score. I'm still not a fan if this "dual submission" of method and benchmark.Hence I will not go any further than this. However, If the CAA methods is accepted and the benchmark not, I would encourage the authors to resubmit at the next venue...

---

### Decision · Program_Chairs · 2024-09-26

**Decision:**

Accept (Poster)

**Comment:**

This paper provides a benchmark for attacks on tabular datasets using the CAA attack, as well as providing robustly trained models for 5 different tasks. Reviewers agree that the paper fills a major gap in the literature, and both the paper and its associated dataset are well-structured and usable. The paper is being accepted but with the very strong recommendation from the reviewers that the paper be updated to include a detailed discussion of CAA, indicating its limitations and that the CAA-based results in the leaderboard be presented with caveats that better attacks may exist